# Numerical Research of Flows into Gullies with Different Outlet Locations

**Rita F. Carvalho** [1,*], **Pedro Lopes** [1], **Jorge Leandro** [2] **and Luis M. David** [3]

1   MARE, Department of Civil Engineering, University of Coimbra, 3030-788 Coimbra, Portugal; pedromiglopes@gmail.com
2   Hydrology and River Basin Management, Technical University of Munich, 80333 München, Germany; jorge.leandro@tum.de
3   National Laboratory for Civil Engineering—LNEC, 1700-066 Lisboa, Portugal; ldavid@lnec.pt
*   Correspondence: ritalmfc@dec.uc.pt

**Abstract:** Gullies are sewer inlets placed in pavements usually covered by bar grates. They are the most common linking-element used to drain a wide range of flows from surface runoff into the buried drainage system. Their hydraulic behavior and their overall hydraulic performance is dependent on the flow conditions, the gully dimension, geometry, and location of the outlet device. Herein a numerical research based on Volume Of Fluid ($VOF$) to detect the interface, and on the Shear Stress Transport $SSTk$-$\omega$ turbulence model was conducted to study the importance of the outlet location and characterize flows through them in drainage conditions. Results provided detailed information about flow features, discharge coefficients, and efficiencies for different outlet locations. The authors identified three different regimes, $R1$, $R2$, and $R3$, and concluded that the outlet location influences the velocity field along the gully, the discharge coefficient, and the drainage efficiency. This allows for the estimation of uncertainty and its variation for different outlet positions.

**Keywords:** drainage systems; gullies; linking-element; discharge capacity; $OpenFOAM^{®}$; $VOF$; $SSTk$-$\omega$ turbulence model

---

## 1. Introduction

In the past decade, there has been an increasing use of dual-drainage models, linking storm water pipe flow (the buried system or minor system) to the overland flow (the major system). Models of the minor system increasingly allow for detailed modelling of the network through representing all manholes, combined sewer overflows, outlet devices, control facilities, among others, usually by defining discharge coefficients [1,2]. On the other hand, some particular structures and devices, such as gullies and manholes, have been analyzed using either experimental investigation [3,4] or computational fluid dynamics (*CFD*) [2] in order to understand flows and search for parameters that characterize such flows. The understanding of the underlying physics and the capability to model these physical phenomena allowing to adequately predict the flow field is of paramount importance to perform numerical environmental studies [5]. This detail has not been extended effectively to the interface devices between the overland drainage system and the buried drainage infrastructure, although it is well known that flooding can be caused or aggravated by insufficient flow capacity of the inlet devices [6–8]. The hydraulic capacity of gullies has been related with factors such as the flow depth on the surface, surface flow conditions, grate type and clogging conditions, inlet area and dimensions, geometry and slope of streets and gullies [8–11]. Factors such as the local geometry and the location of the gully outlet are also known to influence the flow. *CFD* is becoming increasingly used as part of simulation schemes for hydraulic structures in urban drainage systems [2,11–13].

---

In particular, Beg et al. [2] tested different turbulence models and compared them with experimental measurements, concluding that Shear Stress Transport (*SST*) *k-ω* and Renormalization Group (*RNG*) *k-ϵ* turbulence models reached best accuracy. Also Lopes et al. [13] presented simulations of a specific gully using *SSTk-ω* with a high accuracy.

　　Different gully geometries may be found throughout the world. It seems that their geometry and details follow some regional traditional criteria and that sometimes they are constrained by the street and sewer system location. Figure 1 illustrates two gullies, where the connections to the buried system are vertical and located in two different positions along the gully longitudinal axis. The inlet efficiency of gullies is not well known, and their discharge coefficients depend mainly on the geometry, the hydraulic head, and the velocity field [14]. In that work, the authors found that the initial conditions have a great influence in the time to attain steady conditions. They concluded that the outlet position can influence discharge coefficients. This work aims to study the hydraulic behavior of gullies in usual drainage conditions featuring different locations of the vertical outlet pipe that connects the gully to the buried drainage system, taking into consideration different outlet dimensions, as well as a large range of discharge flows, and 2D and 3D analyses.

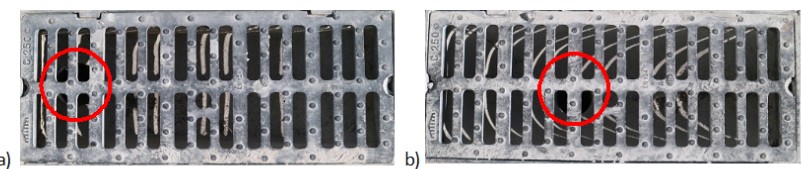

**Figure 1.** Two up-views of two different typical gullies with different location of the gully outlet (connection to the pipe system): (**a**) upstream; (**b**) near center (red circles indicate the gully outlet position).

　　The 3*D* model constructed using *OpenFOAM*<sup>®</sup> code was validated in [10,11,15,16], and [13] by comparing with experimental measurements. Different gully models for different outlet locations and dimensions were now constructed and those were used to simulate a wide range of discharge flows. We present the qualitative description of the gully flow in steady and unsteady conditions as well as the quantitative analysis of the following parameters along the gully: water depths, streamlines, velocity, pressure fields, and the inlet discharge capacity.

## 2. Numerical Model

　　The numerical model is based on the Navier–Stokes equations/Reynolds-Averaged Navier–Stokes (*RANS*) equations governing the motion of the 3*D* incompressible and isothermal flows in which the free surface is described using a Volume-Of-Fluid method (*VOF*). According to this description [17,18], the *VOF*-function, $F = F(x, y, z, t)$, ranging from 0 to 1 and corresponding respectively to cells without water and full occupied by water, is included in the mass and momentum conservation equations. It is also updated using an advection equation for *F*. Some improvements of *VOF* models were developed including surface tension and the interface curvature, as well as the artificial compression of the interface, to improve accuracy of the interface [19]. The *VOF* method used in the interFoam solver implemented in the *OpenFOAM*<sup>®</sup> (Equations (1)–(3)) has two particularities: a volumetric surface force, explicitly estimated by the Continuum Surface Force (*CSF*) function of the surface tension, and the interface curvature, which are included in the momentum equation [20]; the compression of the interface is achieved by introducing an extra, artificial compression term in the advection equation [19,21].

$$\nabla \cdot \mathbf{\bar{u}} = 0 \tag{1}$$

$$\frac{\partial \rho \mathbf{\bar{u}}}{\partial t} + \nabla \cdot (\rho \mathbf{\bar{u}}\mathbf{\bar{u}}) = -\nabla p^* - \mathbf{g} \cdot \mathbf{x}\nabla \rho + \nabla \cdot \boldsymbol{\tau} + \mathbf{f} \tag{2}$$

$$\frac{\partial \alpha}{\partial t} + \nabla \cdot (\alpha \bar{\mathbf{u}}) + \nabla \cdot [\bar{\mathbf{u}}_c \alpha (1 - \alpha)] = 0 \tag{3}$$

where $\bar{\mathbf{u}}$ is the mean velocity vector, $p*$ is the modified pressure adapted by removing the hydrostatic pressure from the total pressure, $\alpha$ is the $VOF\ F$ function, $t$ is the time, $\rho$ is the fluid density, $\mathbf{g}$ is the acceleration due to gravity, $\boldsymbol{\tau}$ is the shear stress tensor, $\mathbf{f}$ is the volumetric surface tension force (where $CSF$ and interface curvature are included) and $\bar{\mathbf{u}}_c$ is the compression velocity.

Further in the mass and momentum conservation equations, $VOF$-function is included through physical properties such as density and viscosity, which are defined by a weighting of the values for air and water. Lopes et al. [13] developed an air-entrainment model. The new solver, airInterFoam, considers the air entrainment triggered by an additional advection equation for dispersed gas phase. Lopes et al. [13] thus simulated a gully with air and water flow, led to conclusion on whether to consider the air-entrainment or not. Although important air-entrainment occurs, it revealed small influence on the hydraulic performance of the gully. They also used a turbulence model, where turbulence variables were calculated with the $SSTk$-$\omega$ turbulence model. This is known for the best combination of two Reynolds-Averaged Simulation ($RAS$) formulations, using high-Reynolds-number formulation of k-$\epsilon$ model for the free-stream region and taking advantage of the accuracy and robustness of $k$-$\omega$ model in the near-wall zone. This study follows the Lopes et al. [13] methodology. However, the air-entrainment model was not used as it is not needed for the detailed requirements, saving computer time.

Total Variation Diminishing ($TVD$) limited form of central-differencing is used for convective terms in momentum equation. The Van-Leer scheme is used for the convective term in $VOF$-advection equation and 'Interface Compression' scheme is used in order to bound the solution of the compressive term between 0 and 1. To ensure boundedness of the phase fraction and avoid interface smearing, the solution of the $VOF$ equation is done with the Multidimensional Universal Limiter for Explicit Solutions ($MULES$). The Pressure-Implicit with Splitting of Operators ($PISO$) procedure proposed by [22] is used for pressure velocity coupling in transient calculations with 3 loops. We used the same discretization schemes employed in previous works on gullies tested with positive outcome [10,13] as well as we used turbulence models with the best accuracy [2].

The computational domain of the flow, aiming to represent a length (L) $\times$ width $\times$ height = 0.6 m $\times$ 0.3 m $\times$ 0.3 m gully placed in a 1% sloped piece of the street pavement, was defined by a box 3.3 m long (0.0 m $< x <$ 3.3 m), 0.5 m wide ($-0.25$ m $< y <$ 0.25 m) and 1.4 m high ($-0.3$ m $< z <$ 1.1 m), using a grid of cells with variable spacing (0.01 m minimum).

The gully was placed between 1.2 m and 1.8 m of the street (1.2 m $< x <$ 1.8 m). The mesh analysis was done and presented in [13]. We tested three meshes. The two finer meshes showed equivalent vortex details, therefore we choose an intermediate mesh size, which retains the main features of the vortices, while also keeping the calculation time acceptable. We constructed the $3D$ geometry using blockMesh utility from $OpenFOAM^{\circledR}$ adapting the work of [10,13]. Figure 2 presents the longitudinal cross sections of the eight gully configurations studied indicating the outlet vertical pipe position center ($P_{op}$). We consider five configurations for an outlet pipe of diameter $D_{op}$ = 80 mm (Configurations 1 to 5, which corresponds to five different locations from upstream to downstream) and three configurations for an outlet pipe of diameter $D_{op}$ = 200 mm (Configurations 6 to 8, which corresponds to three different locations from upstream to downstream). Table 1 summarizes the outlet pipe characteristics for each configuration.

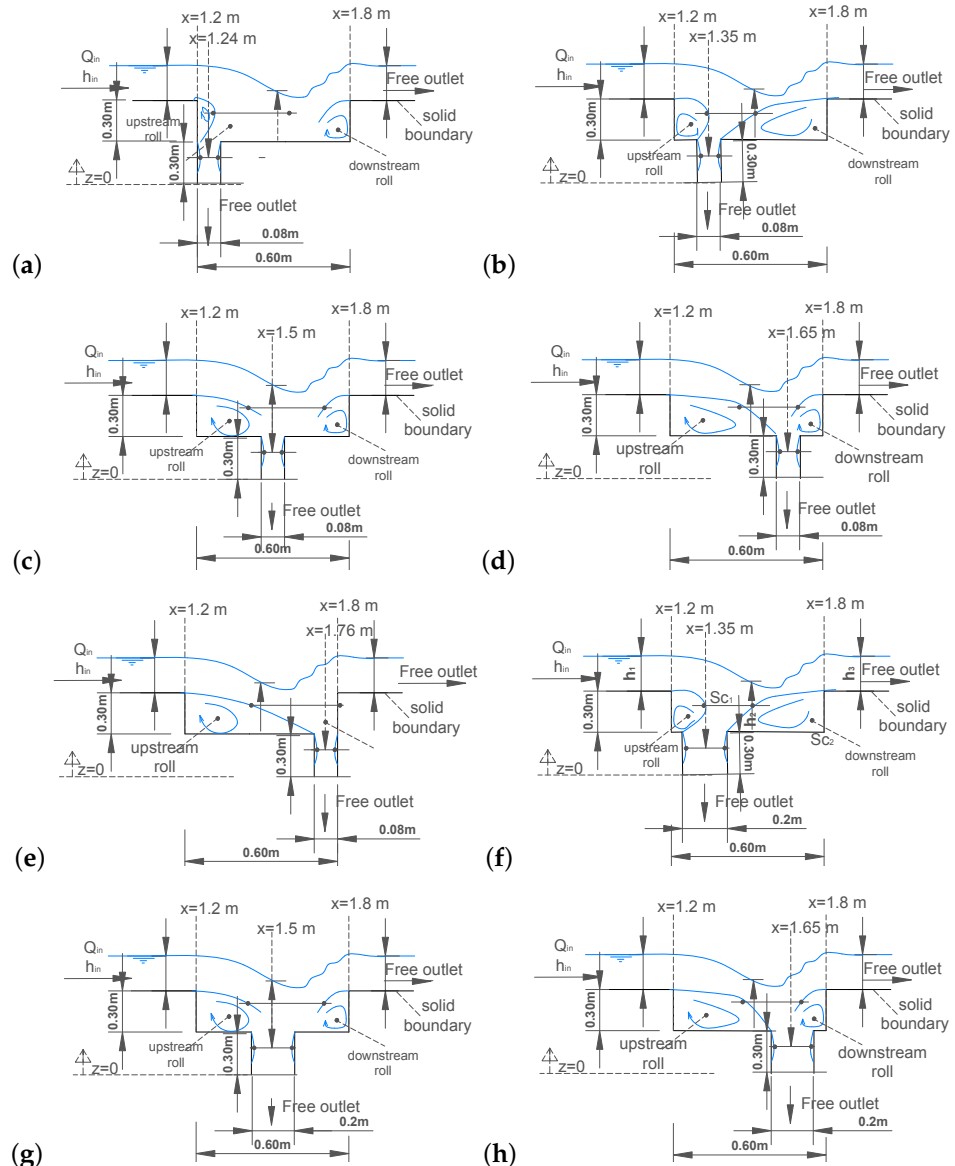

**Figure 2.** Gully longitudinal profiles for the eight vertical outlet pipe configurations: (**a**) Config. 1—$D_{op}$ = 80 mm, $P_{op}$ = 1.24 m; (**b**) Config. 2—$D_{op}$ = 80 mm, $P_{op}$ = 1.35 m; (**c**) Config. 3—$D_{op}$ = 80 mm, $P_{op}$ = 1.5 m; (**d**) Config. 4—$D_{op}$ = 80 mm, $P_{op}$ = 1.65 m; (**e**) Config. 5—$D_{op}$ = 80 mm, $P_{op}$ = 1.76 m; (**f**) Config. 6—$D_{op}$ = 200 mm, $P_{op}$ = 1.35 m; (**g**) Config. 7—$D_{op}$ = 200 mm, $P_{op}$ = 1.5 m; (**h**) Config. 8—$D_{op}$ = 200 mm, $P_{op}$ = 1.65 m.

**Table 1.** Outlet pipe characteristics for the studied configurations.

| Configuration | Figure | $D_{op}$ (mm) | $P_{op}$ (m) | Location |
|---|---|---|---|---|
| 1 | Figure 2a | 80 | 1.24 | upstream edge |
| 2 | Figure 2b | 80 | 1.35 | 1/4 $L$ |
| 3 | Figure 2c | 80 | 1.50 | center |
| 4 | Figure 2d | 80 | 1.65 | 3/4 $L$ |
| 5 | Figure 2e | 80 | 1.76 | downstream edge |
| 6 | Figure 2f | 200 | 1.35 | 1/4 $L$ |
| 7 | Figure 2g | 200 | 1.50 | center |
| 8 | Figure 2h | 200 | 1.65 | 3/4 $L$ |

To simplify the mesh construction, we considered xx-axis parallel to the channel bottom instead of it being horizontal. Thus, the acceleration due to gravity, instead of being considered vertical, was set to $g = (0.00017; 0; -9.81200)$ to consider a 1% slope. The different configurations of the gully outlet were simulated for different flow rates (3 discharges for Config. 1 to 5: Q10, Q20, and Q50) and 4 discharges for Config.6 to 8: Q20, Q50, Q100, and Q200).

The upstream inflow boundary condition (BC) defined as Dirichlet-BC, at $x = 0$, was set according to supercritical uniform conditions in a hypothetical 0.5 m wide channel and 1% slope street. Table 2 presents the inflow conditions for the different discharge flows in supercritical uniform conditions, supercritical uniform height, $h_{in}$, and uniform velocity, $U_{in}$, which was considered constant along the depth, as well as Froude and Reynolds numbers ($Fr = \frac{U_{in}}{\sqrt{gh_{in}}}$, $Re = \frac{U_{in}h_{in}}{\mu}$, with $\mu$ being the viscosity).

**Table 2.** Inflow uniform boundary conditions of numerical investigation program.

| Simulation | $q_{in}$ (L/s/m) | $h_{in}$ (m) | $U_{in}$ (m/s) | $Fr$ (-) | $Re$ (-) |
|---|---|---|---|---|---|
| Q10 | 20 | 0.027 | 0.718 | 1.47 | $1.74 \times 10^4$ |
| Q20 | 40 | 0.041 | 0.990 | 1.53 | $3.48 \times 10^4$ |
| Q50 | 100 | 0.075 | 1.343 | 1.57 | $8.70 \times 10^4$ |
| Q100 | 200 | 0.119 | 1.682 | 1.55 | $1.74 \times 10^5$ |
| Q200 | 400 | 0.195 | 2.058 | 1.49 | $3.48 \times 10^5$ |

The inlet and outlet at the right boundary (downstream channel) were defined by a specific gradient for dynamic part of pressure ($p_* = p_{total} - p_{hydrostatic}$). In the outlet at the bottom (outlet pipe) a hydrostatic pressure was assumed. The top boundary considered hydrostatic pressure and specific gradient for VOF and velocity. The remain boundaries were considered as walls, imposing zero velocity in the vicinity of the face using Dirichlet-BC.

## 3. Results

All simulations presented in Table 2 were run to understand the flow behavior during surcharge transient conditions. It was found that 15 s is enough to achieve the convergence of the main flow properties in steady conditions (gully and downstream water depth and outlet flows, as well as the total water volume in the domain). However, for low flows, the flow continues to decrease progressively, making it no longer interesting to continue with the simulations.

### 3.1. Free-Surface and Velocity Field

Detailed free-surface and 2D flow velocity pattern in central plane over time is illustrated in Tables 3–5, showing the following three hydraulic behaviors, respectively:

(1) the water volume starts decreasing immediately (Config. 1 to 5 for Q10, Config. 6 to 8 for Q20 and Q50);
(2) the water volume remains relatively stable since the beginning (Config. 1 to 5 for Q20 and Config. 6 to 8 for Q100);
(3) the water volume increases up to a point that the downstream water depth smoothly tends to stabilize; (Config. 1 to 5 for Q50 and Config. 6 to 8 for Q200).

Tables 6 and 7 illustrate 3$D$ velocity field for all simulations under steady conditions for all configurations and discharges. We calculate average results from 15 s $\leq t \leq$ 20 s. It is clear that while for some simulations the flow entering the gully is mostly discharged by the bottom outlet, in other situations the water flows over the gully suggesting that the gully efficiency depends not only on the dimensions but on the location of the outlet pipe. For Q200, Config. 8 ($D_{op} = 200$ mm) presents lower water depth downstream of the gully and for Q100, Config. 7 and 8 ($D_{op} = 200$ mm) present more stable condition than Config. 6.

**Table 3.** Flow through the gully in its central plane—Config. 1 to 5 ($D_{op} = 80$ mm) for Q10 , Config. 6 to 8 ($D_{op} = 200$ mm) for Q50.

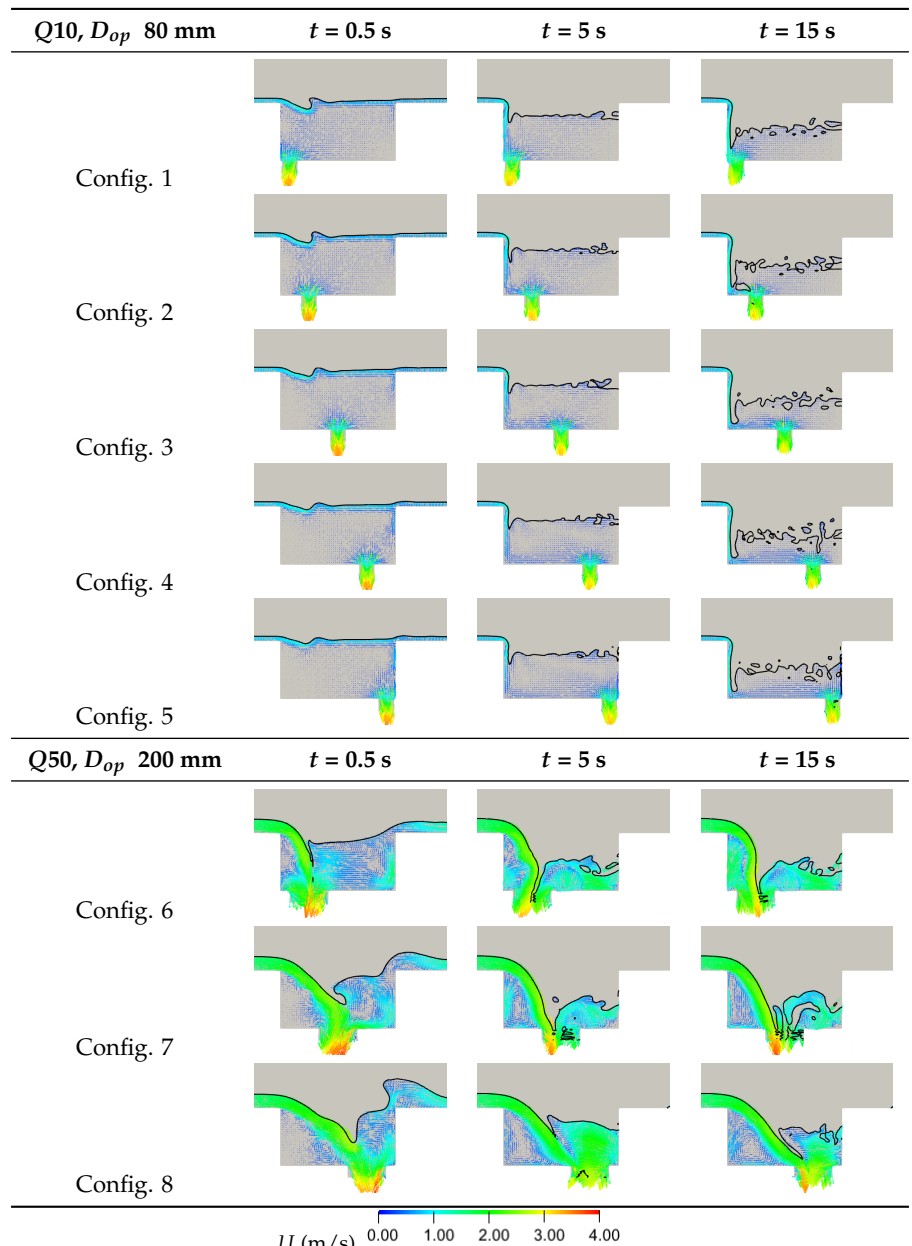

| $Q10, D_{op}$ 80 mm | $t = 0.5$ s | $t = 5$ s | $t = 15$ s |
|---|---|---|---|
| Config. 1 | | | |
| Config. 2 | | | |
| Config. 3 | | | |
| Config. 4 | | | |
| Config. 5 | | | |
| $Q50, D_{op}$ 200 mm | $t = 0.5$ s | $t = 5$ s | $t = 15$ s |
| Config. 6 | | | |
| Config. 7 | | | |
| Config. 8 | | | |

$U$ (m/s)  0.00  1.00  2.00  3.00  4.00 .

**Table 4.** Flow through the gully in its central plane—Config. 1 to 5, $D_{op}$ = 80 mm for Q20 and Configurations 6 to 8, $D_{op}$ = 200 mm for Q100.

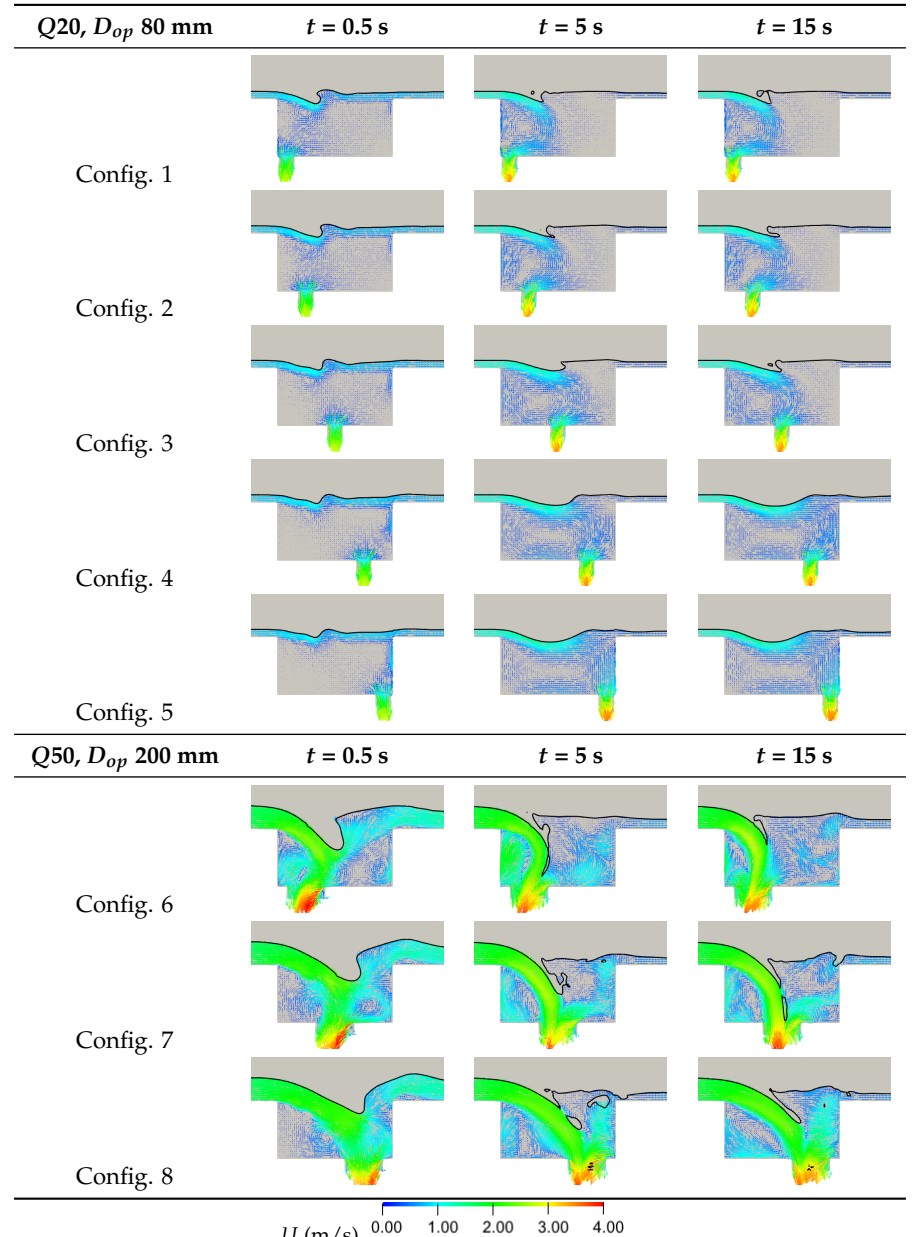

**Table 5.** Flow through the gully in its central plane—Config. 1 to 5, *D* = 80 mm for Q50 and Configurations 6 to 8, *D* = 200 mm Q200.

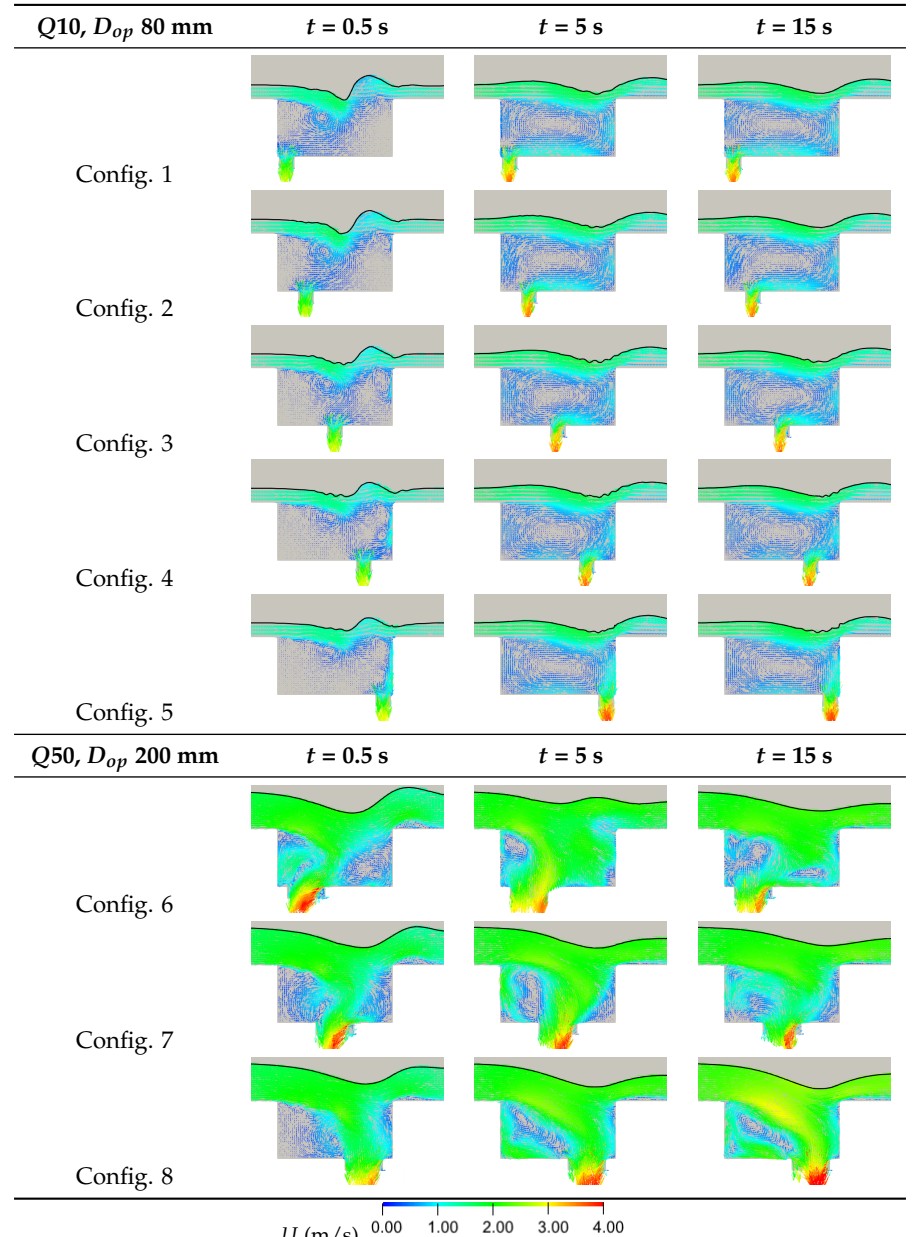

| Q10, $D_{op}$ 80 mm | t = 0.5 s | t = 5 s | t = 15 s |
|---|---|---|---|
| Config. 1 | | | |
| Config. 2 | | | |
| Config. 3 | | | |
| Config. 4 | | | |
| Config. 5 | | | |
| Q50, $D_{op}$ 200 mm | t = 0.5 s | t = 5 s | t = 15 s |
| Config. 6 | | | |
| Config. 7 | | | |
| Config. 8 | | | |

$U$ (m/s)  0.00  1.00  2.00  3.00  4.00

**Table 6.** Velocity field and free-surface configuration at steady conditions—Configurations 1 to 5 (Q10, Q20, and Q50).

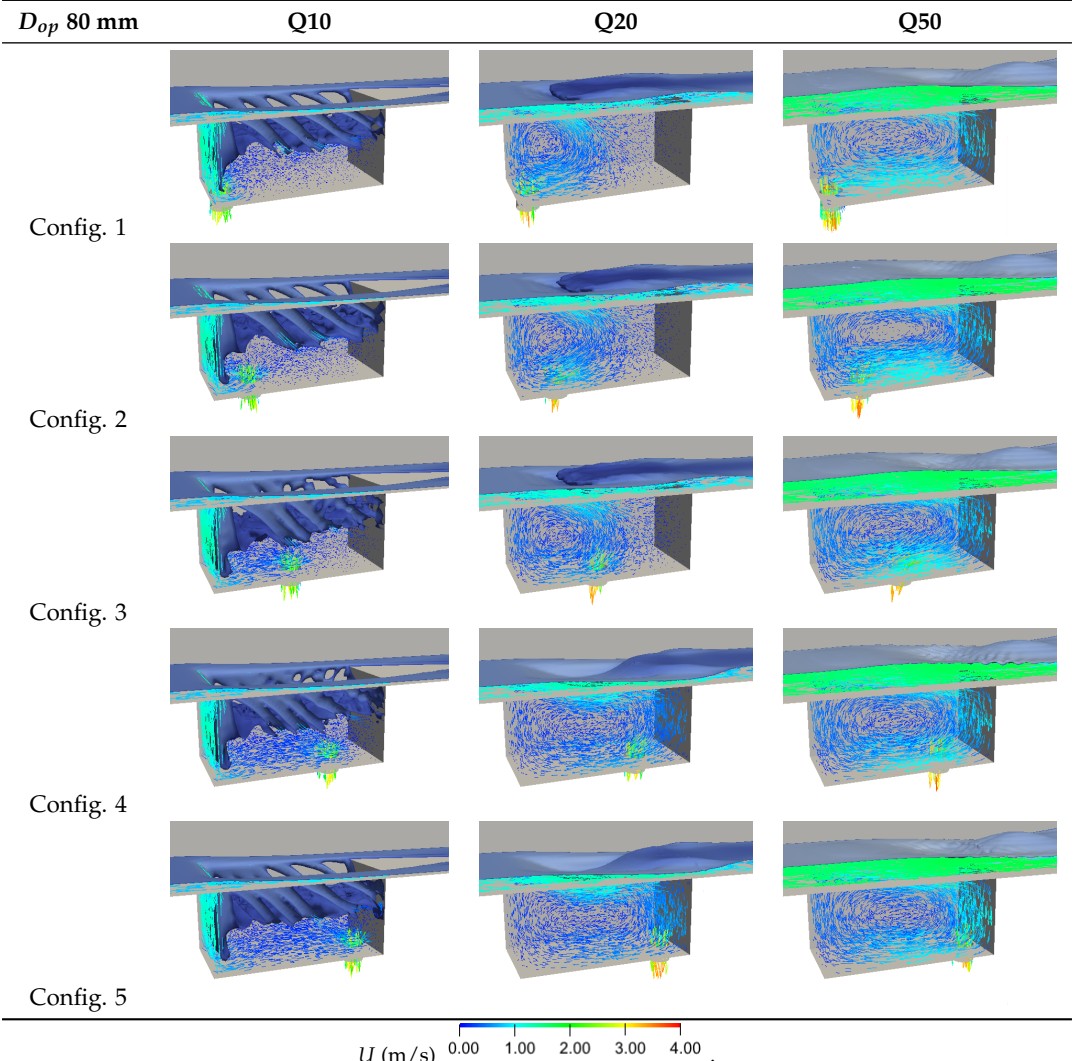

**Table 7.** Velocity field and free-surface configuration at steady conditions—Configurations 6 to 8 (Q20, Q50, Q100, and Q200).

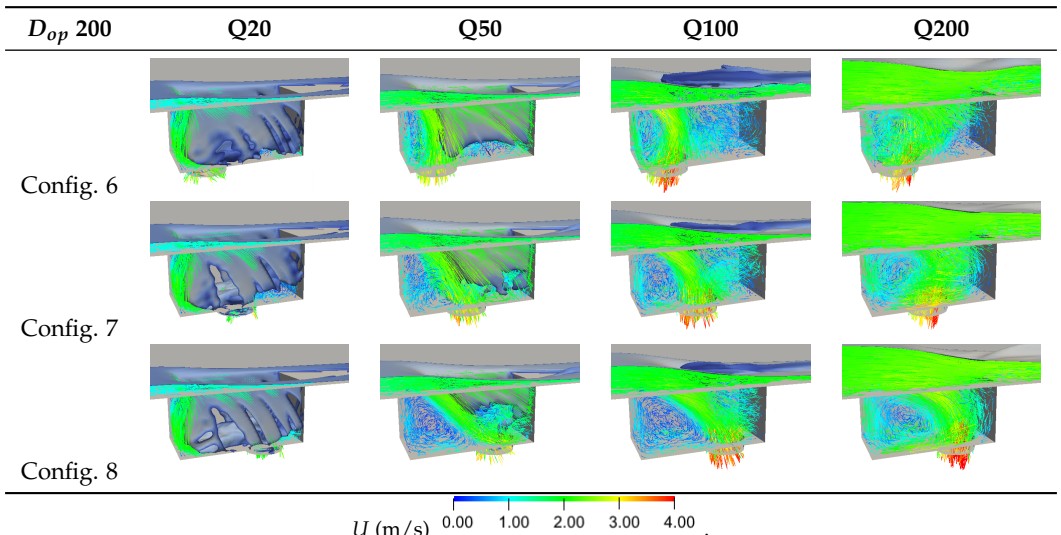

It was found that the natural direction of the jet, as well as the downstream channel water depth can change over time for a constant flow discharge. For Behavior 1, the gully initially ($t$ = 0.5 s) shows to be drowned especially under Config. 7 and 8, and a considerable amount of inflow goes downstream causing large variation of water depth. For Behavior 2, in configurations where the outlet pipe is placed in an upstream or central position (Table 4), it is possible to distinguish 3 phases:

1. at the beginning, the inflow causes a complex velocity field in the gully and the jet tends to reach the gully downstream of the outlet pipe (Table 4, $t$ = 0.5 s);
2. the flow through the outlet pipe drives the jet into the gully bottom outlet controlling the vortices placed upstream and downstream of the outlet pipe. As the main flow is deflected to the outlet pipe, vortices change dimensions (Table 4, $t \leq$ 5 s);
3. further, in almost configurations a hydraulic jump forms at surface above the outlet pipe, also pushing the jet into the outlet pipe. Thus, the discharge capacity increases and the velocity field in the gully downstream stabilizes (Table 4, $t \geq$ 15 s). When the hydraulic jump disappears, it forms a step wave which is responsible for a higher water depth downstream.

While this happens, in Config. 3 and 7, the jet tends to move slightly in the outlet pipe which induce the variation of discharge capacity and the water depth in the gully until equilibrium is attained with oscillating characteristics (Table 4). However, this behavior with oscillatory characteristics cannot be seen in Config. 6 and 8 nor in Config. 1, 2, and 4 to 5. The hydraulic jump occurs for $D_{op}$= 80 mm and Q20 in Config. 1 to 4 and for $D_{op}$= 200 mm and Q100 in all configurations (Config. 6 to 8). However, for Config. 1 to 5 a stable hydraulic jump only occurs for Q20 and Config. 1 to 3. For Behavior 3 (Table 5, simulations for $D_{op}$= 80 mm and Q50, as well as for $D_{op}$= 200 mm and Q200) when the outlet pipe is placed at the center or upstream, the main flow is not deflected to the outlet pipe. In the first seconds, a wave is formed, which increases the water depth downstream. A substantial part of the flow is drained to channel downstream and there is not a hydraulic jump formation. Therefore, the flow through the outlet pipe always comes from the gully downstream area. Similar behavior is observed for simulations from Config. 1 to 5 for Q20 and Q50 and from Config. 6 to 8 for Q100 and Q200.

### 3.2. Velocity and Pressure Profiles

To map the flow and quantify the water depths, velocity, and pressure, values were evaluated at the gully boundaries and top, as well as at the central profile. Figure 3 illustrates velocity and pressure profiles at the gully bottom and top sections ($z$ = 0.3 and $z$ = 0.6 m) for the different configurations and discharge flows. Figure 3 demonstrates that for all the outlet locations with diameter 80 mm, independently of the flow rate, the bottom pressure (Figure 3a), the bottom velocity profile (Figure 3c) and the pressure at the top (Figure 3e) show the same trend. However, while for discharge Q10, the flow at the outlet pipe is drained through upstream side of the bottom outlet pipe, for Q20 and Q50, the flow is drained from downstream side to the outlet pipe (maximum velocity at the right).

At the top entrance (Figure 3g), different patterns can be detected for the velocity longitudinal profile:

(i) outlet pipe located upstream with lower discharges presenting velocity into the gully in upstream area and smaller velocity variation along the gully;
(ii) outlet pipe located in center or downstream with larger discharges presenting velocity into the gully in upstream area and velocity out of the gully in the gully center; and
(iii) outlet pipe with larger discharges presenting small velocity upstream, as well as the highest values for velocity downstream.

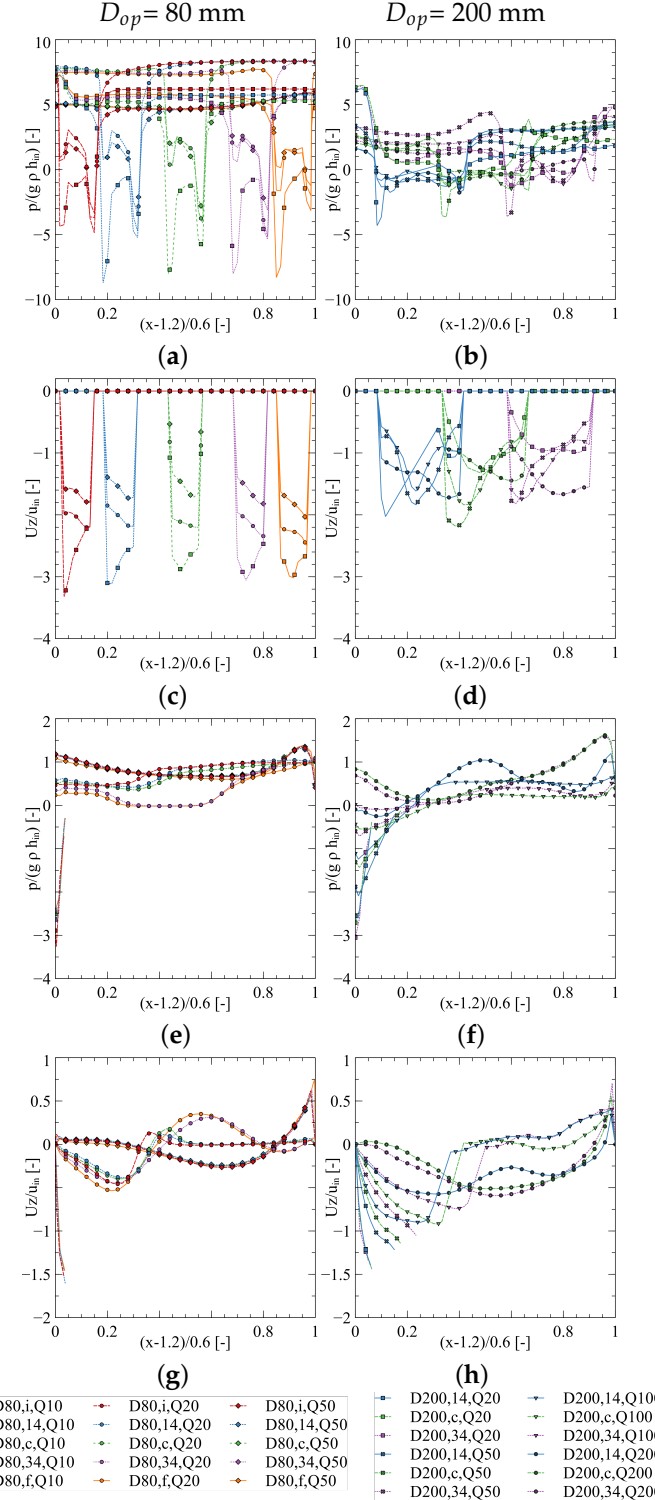

**Figure 3.** Pressure (**a,b,e,f**) and velocity (**c,d,g,h**) profiles at the gully bottom (*z* = 0.3 m)—(**a–d**) and top (*z* = 0.6)—(**e–h**) for *D* = 80 mm (**a,c,e,g**) and *D* = 200 mm (**b,d,f,h**).

Considering all the results, it is clear that the 200 mm outlet pipe in the different locations (Figure 3b,d,f,h) present larger variations along the gully than the 80 mm outlet pipe in the different locations. Pressure values at the gully bottom are larger downstream the outlet for the largest discharges. Different discharges induce different maximum velocity locations into the outlet pipe and different velocity profiles for each location and discharge. The flow in higher discharges for each outlet pipe diameter comes from downstream, similarly in $D_{op}$ = 80 mm and $D_{op}$ = 200 mm outlets. Figure 3f

shows different pressure profiles at the gully top entrance for different locations and discharges. Pressure values on the gully center are higher for larger discharges and upstream outlet pipe location. Higher pressure values at the right wall (large water depth) are presented for larger discharges and location at the center and downstream. On the left (at the gully upstream wall) a separation zone is verified as negative pressure occurs for almost simulation except for higher discharges and outlet pipe location in center or downstream. Figure 3h (velocity profiles at the gully top) shows a very smooth concavity of free surface for the Q200 whereas in the Q100 there is an abrupt rise, which allows the identification of a hydraulic jump occurrence.

## 4. Discussion

### 4.1. Flow Behavior and Characterization

Based on the methodology proposed by [3] and also by [23,24] for drop manholes, we propose the description of transitional flow in a gully organized in 3 different regimes, and based on three aspects, related but independent: (i) the natural jet direction and its relationship with the location of the outlet pipe; (ii) the direction of the streamlines around the outlet pipe location and (iii) the relationship between the discharge in the channel and the outlet pipe discharge capacity. The new regimes which are represented in Figure 4 will be described taking into account the unsteady behavior of the flow in the gully, since the regimes for a specific flow changes according to the water depth in the gully.

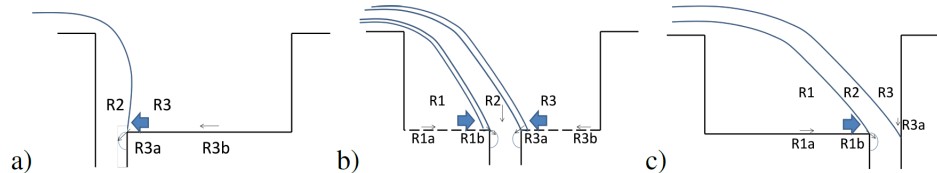

**Figure 4.** Regimes (**a**) R1, (**b**) R2, and (**c**) R3 according each configuration.

*Regime R1 occurs mainly for smaller discharge flows and locations more downstream as the natural jet trajectory tends to reach upstream outlet*—The main flow direction hits the box floor upstream of the bottom outlet pipe, and the streamlines are near horizontal, either along gully bottom in the upstream part, $R1a$, or just near the outlet, $R1b$, causing small discharge capacity. If the discharge in the channel is lower than the bottom outlet pipe discharge capacity, the water depth and the water volume decrease and streamlines follow the bottom wall, and the regime depends only on the jet trajectory, $R1a$ (e.g., Config. 6 and 8, $D_{op}$ = 200 mm for Q20 and Config. 8, $D_{op}$ = 200 mm for Q50, Tables 3 and 7). If the discharge flow in the channel is similar to the outlet capacity, the main flow is driven to the outlet pushed by the vortices located at the gully box upstream part, and it hits the outlet pipe boundary and a contraction is seen, causing asymmetry in the streamlines entering the pipe, $R1b$ (e.g., Config. 7, $D_{op}$ = 200 mm for Q50, Config. 7, $D_{op}$ = 200 mm for Q100, Tables 4, 6 and 7).

*Regime R2 occurs for intermediate discharge flows, when they are similar to the outlet discharge capacity*—The main flow direction hits the outlet pipe, the streamlines are almost symmetric and discharge capacity is maximum. If there is water in the gully, the velocity field in it is influenced by the jet into the outlet pipe and upstream, downstream, or both vortices. This is possibly due to the trend of the jet trajectory or the equilibrium in upstream and downstream vortices (e.g., Config. 5, $D_{op}$ = 80 mm for Q20 and Q50, Config. 6, $D_{op}$ = 200 mm for Q50 and Config. 7, $D_{op}$ = 200 mm for Q50 occasionally and Config. 8, $D_{op}$= 200 mm for Q100, for $t$ 10 s, Tables 3, 4, 6 and 7);

*Regime R3 occurs when the natural jet trajectory overshoots the outlet downstream, for larger discharge flows or outlet pipe gullies located upstream*—The main flow direction tends to hit the area downstream of the outlet pipe, and the streamlines close to it become non-symmetric, which causes a decrease of the discharge capacity. If the outlet pipe capacity is similar or larger than the upstream discharge flow,

the jet streamline is deflected and guided into the outlet pipe. Therefore, a vortex could be created downstream, providing an increase of the discharge capacity, *R3a* (e.g., Config. 1 to 4, $D_{op}$ = 80 mm for Q20; Config. 6, $D_{op}$ = 200 mm for Q100, Config. 6, 7, $D_{op}$ = 200 mm for Q200 and Config. 8, $D_{op}$ = 200 mm for Q200, and for $t \geq 10$ s, Tables 4–7). In this case, during discharge, if the outlet discharge is not enough to discharge the flow from the upstream channel, the water depth and the water volume may initially increase and thus vortices on the gully may be formed deflecting the jet into the outlet pipe, thus, increasing its capacity, *R2* could be attained. Alternatively, if the jet tendency is the gully wall to far downstream from the outlet pipe, the vortices upstream are unable to deflect the flow into the outlet pipe, and thus most of the upstream discharge will flow downstream: *R3b* (e.g., Config. 1 to 4, $D_{op}$ = 80 mm for Q50).

Apart from the regime description, some additional behaviors must be noted from a theoretical standpoint: *R1* and *R2* should not be physically possible on Config. 1; however, *R2* is verified since the flow is forced to curve into the outlet for lower discharges; *R3a* could be defined for Config. 5, since downstream vertical wall forces the flow into the outlet.

### 4.2. Discharge Coefficients

Discharge coefficients, depending on the bottom outlet pipe diameter and water depth inside the gully, can be evaluated considering either orifice (Equation (4)) or weir (Equation (5)) formulas:

$$C_o = \frac{Q_o}{A_o \sqrt{2gH_{in}}} \tag{4}$$

$$C_w = \frac{Q_w}{b_w \sqrt{2gH_{in}^3}}; \tag{5}$$

where subscripts '*o*' and '*w*' represent an orifice and a weir variable, respectively; $Q_o$ and $Q_w$ represent the discharge flow through outlet pipe (either considering orifice or weir), $A$ is the area of the orifice, $b_w$ is the channel width, $g$ is the acceleration due to gravity, and, ultimately, $H_{in}$ is the uniform height of the flow upstream the weir/orifice (in the upstream channel). Efficiency is obtained by calculating the ratio between $Q_o$ or $Q_w$ and $Q_{in}$, which is the sum of $Q_{outlet\_bottom}$ and $Q_{outlet\_channel}$, presented in Tables 8 and 9. Tables 8 and 9, apart discharges, present discharge coefficients considering weir and orifice formulas and bottom outlet efficiency. Figure 5 illustrates the Tables 8 and 9 discharge coefficients and efficiency data.

As expected the higher outlet pipe diameter drain higher flows and largest efficiency is found for lower flows. However, for lower flows and Behavior 1, as defined at the beginning of Section 3 and for each outlet pipe diameter, all discharge coefficients and efficiency are similar. Looking at the coefficients and efficiency values of gullies with different outlet locations, Figure 5, the largest efficiency is found for the center and downstream location (95%). In intermediate discharges, those maintaining volume in the gully (Behavior 2), the higher discharge coefficient and efficiency is also found for the central bottom outlet. For higher discharges, flood conditions, it is found for the downstream position, which performs better. Calculating the differences, 26% is found to be the maximum uncertainty range occurring in the $D_{op}$ = 200 mm and Q200 between Config. 6 and 8 related with Config. 7 (c), followed by 25% for $D_{op}$ = 80 mm and Q50 between configurations 3 and 5 related with Config. 3 (c) and 13% for $D_{op}$ = 200 mm and Q100.

**Table 8.** Discharges, coefficients and efficiency for gullies with different bottom outlet location ($D$ = 80 mm).

| | Config. 1 | Config. 2 | Config. 3 | Config. 4 | Config. 5 |
|---|---|---|---|---|---|
| | $D$ = 80 mm, Q10, $H_{in}$ = 0.027 m | | | | |
| $Q_{outlet\_channel}$ (m$^3$/s) | 0.0097 | 0.0097 | 0.0097 | 0.0091 | 0.0097 |
| $Q_{outlet\_bottom}$ (m$^3$/s) | 0.0091 | 0.0091 | 0.0091 | 0.0090 | 0.0090 |
| $C_w$ | 0.9227 | 0.9215 | 0.9220 | 0.9181 | 0.9180 |
| $C_o$ | 2.4782 | 2.4750 | 2.4763 | 2.4658 | 2.4655 |
| Efficiency (%) | 94% | 93% | 94% | 93% | 93% |
| | $D$ = 80 mm, Q20, $H_{in}$ = 0.041 m | | | | |
| $Q_{outlet\_channel}$ (m$^3$/s) | 0.0092 | 0.0093 | 0.0088 | 0.0091 | 0.0082 |
| $Q_{outlet\_bottom}$ (m$^3$/s) | 0.0111 | 0.0110 | 0.0115 | 0.0112 | 0.0121 |
| $C_w$ | 0.6049 | 0.5964 | 0.6258 | 0.6088 | 0.6595 |
| $C_o$ | 2.4693 | 2.4336 | 2.5542 | 2.4842 | 2.6913 |
| Efficiency (%) | 55% | 54% | 57% | 55% | 60% |
| | $D$ = 80 mm, Q50, $H_{in}$ = 0.074 m | | | | |
| $Q_{outlet\_channel}$ (m$^3$/s) | 0.0397 | 0.0403 | 0.0407 | 0.0401 | 0.0384 |
| $Q_{outlet\_bottom}$ (m$^3$/s) | 0.0107 | 0.0101 | 0.0097 | 0.0103 | 0.0120 |
| $C_w$ | 0.2348 | 0.2220 | 0.2121 | 0.2270 | 0.2646 |
| $C_o$ | 1.7519 | 1.6565 | 1.5828 | 1.6937 | 1.9750 |
| Efficiency (%) | 21% | 20% | 19% | 20% | 24% |

**Table 9.** Discharges, coefficients and efficiency for gullies with different bottom outlet location ($D$ = 200 mm).

| | Config. 6 | Config. 7 | Config. 8 |
|---|---|---|---|
| | $D$ = 200 mm, Q20, $H_{in}$ = 0.041 m | | |
| $Q_{outlet\_channel}$ (m$^3$/s) | 0.0011 | 0.0011 | 0.0011 |
| $Q_{outlet\_bottom}$ (m$^3$/s) | 0.0194 | 0.0194 | 0.0196 |
| $C_w$ | 1.0550 | 1.0552 | 1.0621 |
| $C_o$ | 0.6888 | 0.6889 | 0.6941 |
| Efficiency (%) | 94% | 95% | 95% |
| | $D$ = 200 mm, Q50, $H_{in}$ = 0.075 m | | |
| $Q_{outlet\_channel}$ (m$^3$/s) | 0.0036 | 0.0038 | 0.0041 |
| $Q_{outlet\_bottom}$ (m$^3$/s) | 0.0527 | 0.0530 | 0.0522 |
| $C_w$ | 1.1588 | 1.1656 | 1.1426 |
| $C_o$ | 1.3832 | 1.3911 | 1.3676 |
| Efficiency (%) | 94% | 93% | 93% |
| | $D$ = 200 mm, Q100, $H_{in}$ = 0.119 m | | |
| $Q_{outlet\_channel}$ (m$^3$/s) | 0.0258 | 0.0153 | 0.0183 |
| $Q_{outlet\_bottom}$ (m$^3$/s) | 0.0743 | 0.0853 | 0.0817 |
| $C_w$ | 0.8169 | 0.9381 | 0.8977 |
| $C_o$ | 1.5478 | 1.7772 | 1.7020 |
| Efficiency (%) | 74% | 85% | 82% |
| | $D$ = 200 mm, Q200, $H_{in}$ = 0.195 m | | |
| $Q_{outlet\_channel}$ (m$^3$/s) | 0.1222 | 0.1215 | 0.1015 |
| $Q_{outlet\_bottom}$ (m$^3$/s) | 0.0784 | 0.0796 | 0.0991 |
| $C_w$ | 0.4111 | 0.4175 | 0.5197 |
| $C_o$ | 1.2759 | 1.2957 | 1.6127 |
| Efficiency (%) | 39% | 40% | 49% |

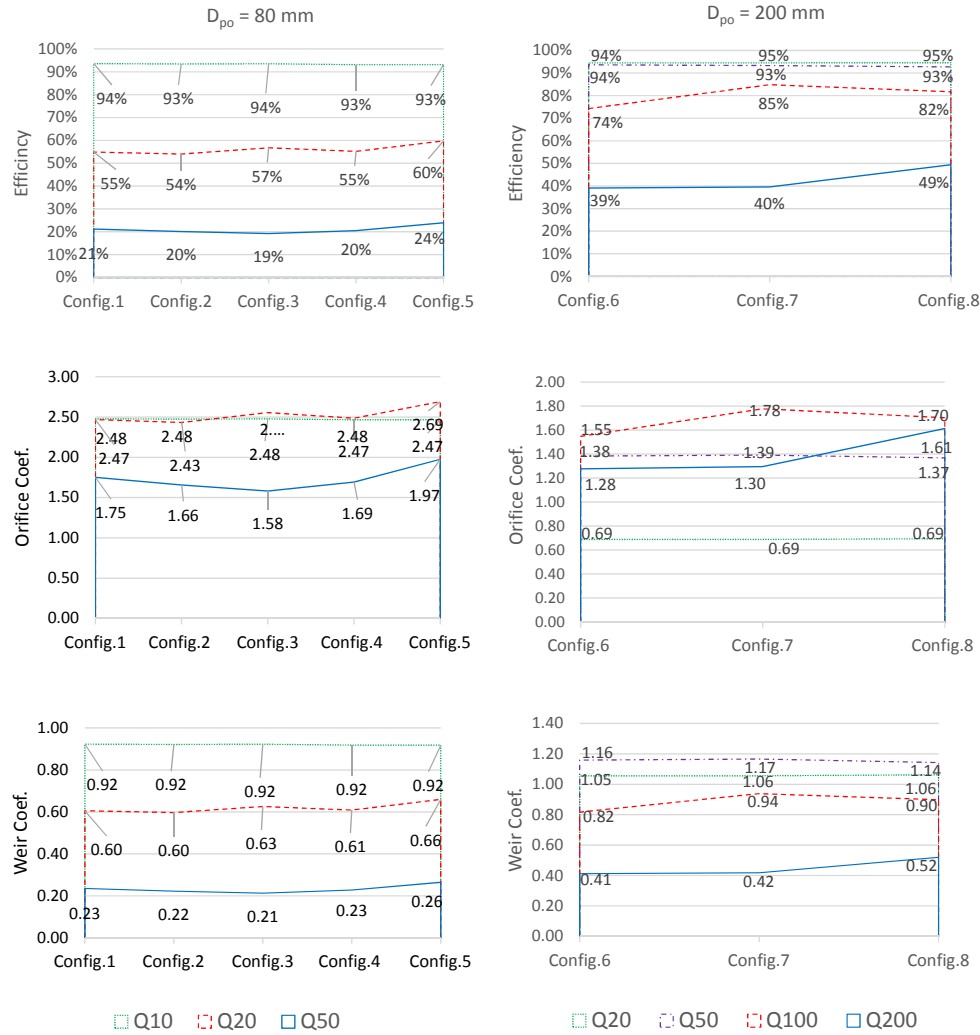

**Figure 5.** Efficiency and Discharge Coefficients maps for $D_{op} = 80$ mm (Config. 1 to 5) and $D_{op} = 200$ mm (Config. 6 to 8).

## 5. Conclusions

An urban drainage gully with $0.6 \times 0.3 \times 0.3$ m$^3$ (length $\times$ width $\times$ height) placed in a channel with 1% slope was modelled to evaluate the importance of the bottom outlet pipe location along the gully longitudinal axis. Five different flow rates and two sizes of the outlet pipe were used in the analysis. Different $3D$ models based on Navier–Stokes/Reynolds equations, $VOF$ method and $SSTk$-$\omega$ were constructed and used in the $OpenFOAM^{\circledR}$ platform, which showed ability to predict complex flow features, air-water interface, and turbulence characteristics. The study was motivated by the increase of applications of dual-drainage models that consider the performance of linking elements between the surface and buried systems, as well as by the need to take uncertainty into account.

The different locations of the bottom outlet pipe lead to different limitation of discharge and influence the stability of the free surface and the water depth along the gully and downstream channel. Three regimes were identified based on the velocity field, which is the main responsible for the hydraulic behavior and also determines the symmetry of the streamlines at the outlet and thus the contraction and the discharge capacity. Besides the initial water depth in the gully, the velocity field and, thus, the regime depend on three main factors: (1) the outlet location; (2) the natural jet direction and the relative distance between the area where it hits the gully bottom and the outlet pipe location, which influence streamlines around the bottom outlet pipe; (3) the relationship between the discharge in the channel and the outlet pipe discharge capacity. The largest difference of 26% was found for the

same outlet diameter and the same upstream flow with different outlet pipe locations corresponding to largest flow and gully pipe bottom outlet diameter (26% for $D_{op}$ = 200 mm and Q200 related with Config. 7, followed by 25% for $D_{op}$ = 80 mm and Q50 related with Config. 3), showing that unknown outlet pipe location creates a significant range uncertainty in flooding conditions.

**Author Contributions:** Conceptualization, R.F.C. and L.M.D.; Formal analysis, R.F.C., P.L., J.L. and L.M.D.; Investigation, R.F.C.; Methodology, R.F.C. and P.L.; Visualization, P.L.; Writing—original draft, R.F.C.; Writing—review and editing, P.L., J.L. and L.M.D.

**Funding:** This study had the support of the FCT (Portuguese Foundation for Science and Technology), through the Project UID/MAR/04292/2019, which was financed by MEC (Portuguese Ministry of Education and Science) and the FSE (European Social Fund), under the programs POPH/QREN (Human Potential Operational Programme from National Strategic Reference Framework) and POCH (Human Capital Operational Programme) from Portugal2020.

**Acknowledgments:** All the numerical results here showed were performed on the Centaurus Cluster of the Laboratory for Advanced Computing of University of Coimbra, Portugal.

**Conflicts of Interest:** The authors declare no conflict of interest.

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
