# Peer review of "Numerical Research of Flows into Gullies with Different Outlet Locations"

_water, doi:10.3390/w11040794_

Round 1

Reviewer 1 Report

This paper, which presents a numerical study of a gulley with two different sized outlets, placed in different longitudinal locations for a range of flow rates, is interesting and seems to provide a useful addition to the field.  The paper is generally well presented and understandable, but I have a number of points which it would be useful to improve.  I am not a CFD expert, so have not commented on the CFD parts of the methodology.  I have also attached a pdf which includes more minor typos and suggestions to improve some minor English issues.

Page 2, lines 7 to 9 - please revisit this sentence, the meaning is not clear.

Page 2, line 21 and throughout - author names in references are missing spaces - e.g. 'Begetal.' should be 'Beg et al.'

Page 4, line 70 - should 'Total Variation Dimishing' be 'Total Variation Diminishing'?

Page 4, line 76 - should  'Pressure-Implicit with Spitting of Operations' be 'Pressure-Implicit with Splitting of Operations '

Page 4, lines 83 to 85 - I don't think all of these variables be x?

Tables 2 to 6 - a colour bar is included, but the legend is too small to read.  Could a single colour bar for each table at a legible size be used?  Also each individual figure is very small, could these be made larger (e.g. 2x), maybe in supplementary material?  Also for Tables 2 to 4 consider adding the i, c, f, etc suffixes used in later tables.

Page 14 - this later part of section 3.1 could be expanded to be clearer, including better reference to the tables, as is done earlier in this section.

Page 14, line 174 - Consider replacing 'mainly' with 'not only' in '...gully efficiency depends mainly on the dimensions but ...'. 
Figure 3 - the legend is illegible when printed and the axis titles only just big enough to read, it's also difficult to distinguish between the marker shapes even when viewing a zoomed in pdf.  Please consider revising this useful Figure to be easier to read, one option might be to use a single legend for the whole figure so a larger font can be used and also make the markers larger.

Page 19-20, the descriptions of the regimes are very difficult to follow because of long sentences.  Please consider splitting these up to improve readability.

Page 20, line 271, start a new paragraph for the sentence beginning 'Apart of ..'.  Also change 'of' to 'from'.

Page 21, line 281 - the weir is not defined - is it the perimeter of the gulley?

Table 7 and 8 - Font for the word 'Efficiency' varies.

Page 24, line 290-291 - '... for each pipe outlet diameter, all discharge coefficients
and efficiency are similar.' this does not seem to be the case for Dpo 200 at Q20 and Q50?

Figure 5 - As with other Figures, can this be made slightly larger to be legible in print form?  Also, top left y axis 'Efficiency' is spelt wrong, bottom two rows, po in Dpo is not subscripted.

Page 26, lines 317 - I might have misunderstood something, but item 2 doesn't seem to be covered by the results presented?  For any single case fo flow / configuration only one initial water depth in the gulley is considered?

Page 26, line 319 - Is point 4 effectively the same as point 1 for the results presented?  If more gulley sizes were investigated, point 4 might apply.

References - please check the content and formatting, some are missing journal titles or other details (e.g. 9, 19, 24) and many have the year in twice.

Author Response

Dear Editor and Reviewers,

We thank you for your fruitful comments that clearly improved the paper. An English expert reviewed the paper. Apart from a certain number of commas,  all the changes are in blue.

Please find answers, comments and modifications above each reviewer’s issue.

Comments and Suggestions for Authors

This paper, which presents a numerical study of a gulley with two different sized outlets, placed in different longitudinal locations for a range of flow rates, is interesting and seems to provide a useful addition to the field.  The paper is generally well presented and understandable, but I have a number of points which it would be useful to improve.  I am not a CFD expert, so have not commented on the CFD parts of the methodology.  I have also attached a pdf which includes more minor typos and suggestions to improve some minor English issues.

Page 2, lines 7 to 9 - please revisit this sentence, the meaning is not clear.

Sentence revised, we rephrased the sentence: On the other hand, some flow analysis in particular structures and devices, such as gullies, and manholes, search detailed characteristics were analysed using either experimental investigation {3,4} or CFD (2), in order to understand flow behaviour and look for parameters that could characterise such flows.

Page 2, line 21 and throughout - author names in references are missing spaces - e.g. 'Begetal.' should be 'Beg et al.'

Corrected.

Page 4, line 70 - should 'Total Variation Dimishing' be 'Total Variation Diminishing'?

Corrected.

Page 4, line 76 - should  'Pressure-Implicit with Spitting of Operations' be 'Pressure-Implicit with Splitting of Operations '

Corrected: Pressure-Implicit with Splitting of Operators (Issa, 1986).

Page 4, lines 83 to 85 - I don't think all of these variables be x?

Corrected: x, y and z.

Tables 2 to 6 - a colour bar is included, but the legend is too small to read.  Could a single colour bar for each table at a legible size be used?  Also each individual figure is very small, could these be made larger (e.g. 2x), maybe in supplementary material?  Also for Tables 2 to 4 consider adding the i, c, f, etc suffixes used in later tables.

The tables 2 to 6 was renamed as Tables 3 to 7 and all figures were reformulated and implemented in a corrected way to not loose definition. We present now just one legend valid for all sub Figures, which are fewer and larger. Adding the suffixes i c and f is against opinion from other reviewer. We decided to take off the suffixes and present Figures in a simpler way with just the name of the configuration. Figures are now less crowded as suggested by the reviewers.

Page 14 - this later part of section 3.1 could be expanded to be clearer, including better reference to the tables, as is done earlier in this section.

Section 3.1 and 3.2 were reformulated into one to present results section more concise as suggested by the reviewers.

Page 14, line 174 - Consider replacing 'mainly' with 'not only' in '...gully efficiency depends mainly on the dimensions but ...'. 

Replaced.

Figure 3 - the legend is illegible when printed and the axis titles only just big enough to read, it's also difficult to distinguish between the marker shapes even when viewing a zoomed in pdf.  Please consider revising this useful Figure to be easier to read, one option might be to use a single legend for the whole figure so a larger font can be used and also make the markers larger.

We made a new Figure 3 no dimensional, with better quality and with just one legend, valid for all. Axis titles are with larger font

Page 19-20, the descriptions of the regimes are very difficult to follow because of long sentences.  Please consider splitting these up to improve readability.

This section was re-written.

Page 20, line 271, start a new paragraph for the sentence beginning 'Apart of ..'.  Also change 'of' to 'from'.

Corrected.

Page 21, line 281 - the weir is not defined - is it the perimeter of the gulley?

We considered bw as the channel width and we added this in the tex.

Table 7 and 8 - Font for the word 'Efficiency' varies.

Corrected.

Page 24, line 290-291 - '... for each pipe outlet diameter, all discharge coefficients
and efficiency are similar.' this does not seem to be the case
for Dpo 200 at Q20 and Q50?

All the coefficients were verified. This sentence was related to the behaviour 1. Text was re-written.

Figure 5 - As with other Figures, can this be made slightly larger to be legible in print form?  Also, top left y axis 'Efficiency' is spelt wrong, bottom two rows, po in Dpo is not subscripted.

Figure was corrected. Fonts for axis and legend are now larger. Efficiency was corrected and po is now subscript.

Page 26, lines 317 - I might have misunderstood something, but item 2 doesn't seem to be covered by the results presented?  For any single case fo flow / configuration only one initial water depth in the gulley is considered? Page 26, line 319 - Is point 4 effectively the same as point 1 for the results presented?  If more gulley sizes were investigated, point 4 might apply.

Text was re-written to clarify this issue.

References - please check the content and formatting, some are missing journal titles or other details (e.g. 9, 19, 24) and many have the year in twice.

We include missing data and remove year, when twice.

Reviewer 2 Report

This paper presents a numerical simulation study of flow into gullies with different outlet locations. The authors identify three flow regimes, depending on the flow configuration, and discuss their impact on discharge coefficient and bottom outlet efficiency.

I think this work can be useful in the design of dual-drainage systems. It is also an interesting practical application of computational fluid dynamics in engineering design. The authors have used a similar numerical model in several other publications, so the computational approach does not seem to be significantly new (Lopes et al. J. Irrig. Drain. Eng. 2016; Leandro et al. Computers & Fluids 2014; Lopes et al. Water Science & Technology 2017; Martins et al. Water Science & Technology 2014).

The main weaknesses of the work, in my opinion, are the following:

- The discussion is mostly qualitative. The authors describe what they see in the simulations, but there is no a clear insight into flow processes that would allow to generalize their observations to other cases (say, as in an analysis based on dimensionless quantities like the Reynolds number).

- The quality of the figures is very poor. In general, there is way too much information (as in Figures 2 and 3). The resolution is also poor, so that zooming-in doesn’t help. This applies to Tables 2-6 in particular: it’s hard to see what’s being plotted, the color bars are blurry etc.

- The writing is often confusing. The description of flow regimes in the Discussion, for example. Each regime is described using a set of short statements, separated by semicolons, as if they were taken from bullet points in a draft. And then the authors indicate which configurations lead to those regimes using complicated references to simulation cases that include parenthesis inside another parenthesis.

In summary, I think this work deserves to be published, but only after a major revision that significantly improves the quality of the presentation. I understand that there are many simulation cases, but the authors need to find a way to organize the material so that it can be properly read and understood. 

Detailed comments: 

- This is a numerical study. What is the influence of mesh size on these results?

- Need to rewrite lines 88 to 104. This description is very difficult to follow.  It’s a confusing paragraph that describes a crowded figure (Fig.2). Somehow the authors need to find a way to simplify the description of their flow configurations. Maybe split Fig.2 into two figures.

- Figure 2. There’s just too much information in this figure. I found it really difficult to understand what is being described. In the main text there’s a confusing reference to Config. 1, up to Config. 8, and then things like “initialposition(i)”,  “Ppo= 1.35m(1/4)” , “centre(c)” “center(c)” “finalposition(f)” etc.

- Table 2-4 is more a figure than a table. The quality of the plots is very poor, so that it’s impossible to see anything beyond simple colors and basic flow patterns. Does the black contour represent the air-water interface?

- Section 3.1. The description of Tables 2 and 3 is qualitative, and there are too many configurations and panels to discuss. Do we learn anything in terms fundamental flow processes?

- Figure 3. The quality is again very poor. Is there a way to simplify all this information, so that the figure conveys the main message?

Other minor comments:

-Lines 21, 24. Formatting of citations.

- Simulatiions, line 108.

- Lines 114-116 needs to be rewritten: “The outlet boundary at right (street outlet) was set as free outflow, considering outlet zero gradient for all except for the pressure which dynamic part is specified as zero normal gradient rather than total”

- Line 118, “hear”

- Line 119, what does “exchanges of air” mean as a boundary condition? Imposing atmospheric pressure?

- Use of italics. Units and other letters should be converted to regular text in latex, with \text{} or some other command in math mode.

- Line 124. I’m surprised that, given the flow complexity and presumably large Reynolds number, the numerical solutions converge to a steady state. Is this because the grid is very coarse?

- Line 128 needs to be rewritten “the water volume maintains relatively stable since the beginning”

- Line 137. “Particullary”

- Line 150. “Responsable”

- Lines 162-164. This phrase should be rewritten, as it is difficult to read: “Therefore, in the first seconds a wave is formed increasing water depth downstream, a substantial part of the flow is drained to channel downstream and the flow through the pipe outlet always come from the gully downstream area”

- Lines 175-177. The use of parenthesis inside another parenthesis is very confusing.

- Lines 185. “Almost similar” is too vague.

- Lines 185-188, the use of commas seems incorrect.

- Lines 214-216 should be rewritten (maybe split into two sentences?).

- Lines 216-220 should to be rewritten (we need at least a few commas).

- Figure 4, caption. Maybe “according to each configuration”?

- Line 230 “…and the location of the pipe outlet.”

- Lines 233-245. This reads like a one-sentence paragraph that needs to be split into several sentences. The use of semicolons seems odd.

- Lines 247-254, same as for lines 233-245.

-  Lines 256-271, same as for lines 233-245.

Author Response

Dear Editor and Reviewer 2,

We thank you for your fruitful comments that clearly improved the paper. An English expert reviewed the paper. Apart from a certain number of commas,  all the changes are in blue.

Please find answers, comments and modifications above each reviewer’s issue.

Comments and Suggestions for Authors

This paper presents a numerical simulation study of flow into gullies with different outlet locations. The authors identify three flow regimes, depending on the flow configuration, and discuss their impact on discharge coefficient and bottom outlet efficiency. I think this work can be useful in the design of dual-drainage systems. It is also an interesting practical application of computational fluid dynamics in engineering design. The authors have used a similar numerical model in several other publications, so the computational approach does not seem to be significantly new (Lopes et al. J. Irrig. Drain. Eng. 2016; Leandro et al. Computers & Fluids 2014; Lopes et al. Water Science & Technology 2017; Martins et al. Water Science & Technology 2014).

These mentioned works focus validation of numerical works against experimental data in a specific gully. In this work, we studied the influence of outlet pipe location in the gully fluid dynamics. This was highlight in the text.

The main weaknesses of the work, in my opinion, are the following:

- The discussion is mostly qualitative. The authors describe what they see in the simulations, but there is no a clear insight into flow processes that would allow to generalize their observations to other cases (say, as in an analysis based on dimensionless quantities like the Reynolds number).

Fluid dynamics in the gully are described qualitatively to understand flow behaviour through the gully with different outlet pipe location (vortices formation and location, which influence the flow). This description support the definition of three different regimes. Different pressure, velocity and discharge coefficients values represent quantitative description. This was highlight in the text in the final part of introduction.

- The quality of the figures is very poor. In general, there is way too much information (as in Figures 2 and 3). The resolution is also poor, so that zooming-in doesn’t help. This applies to Tables 2-6 in particular: it’s hard to see what’s being plotted, the color bars are blurry etc.

The quality of the figures in the text improved substantially. They were of good quality but implemented in a wrong way. The figures were simplified and are now less crowded.

- The writing is often confusing. The description of flow regimes in the Discussion, for example. Each regime is described using a set of short statements, separated by semicolons, as if they were taken from bullet points in a draft. And then the authors indicate which configurations lead to those regimes using complicated references to simulation cases that include parenthesis inside another parenthesis.

The text describing flow regimes was reviewed, simplified and re-organized.

In summary, I think this work deserves to be published, but only after a major revision that significantly improves the quality of the presentation. I understand that there are many simulation cases, but the authors need to find a way to organize the material so that it can be properly read and understood. 

We made a major revision, we reviewed all the text, we reduce the number of the presented simulation results.

Detailed comments: 

- This is a numerical study. What is the influence of mesh size on these results?

We made a mesh study for gullies testing three meshes and the study is presented in other papers. We add in the text: The two finer meshes showed equivalent vortex details, therefore we chose an intermediate mesh size, which retains the main features of the vortices, while keeping the calculation time acceptable.

- Need to rewrite lines 88 to 104. This description is very difficult to follow.  It’s a confusing paragraph that describes a crowded figure (Fig.2). Somehow the authors need to find a way to simplify the description of their flow configurations. Maybe split Fig.2 into two figures. There’s just too much information in this figure. I found it really difficult to understand what is being described. In the main text there’s a confusing reference to Config. 1, up to Config. 8, and then things like “initialposition(i)”,  “Ppo= 1.35m(1/4)” , “centre(c)” “center(c)” “finalposition(f)” etc.

Figure 2 description was simplified and a new table, as suggested for reviewer 3, was added to simplify description. Figure 2 was simplified and sized.

- Table 2-4 is more a figure than a table. The quality of the plots is very poor, so that it’s impossible to see anything beyond simple colors and basic flow patterns. Does the black contour represent the air-water interface?

Table 2-4 (now tables 3 to 5) were reorganised. Figures (now less and larger) were implemented correctly without losing definition. Just a legend valid for all is included. The black contour does represent the air-water interface and is included in the text.

- Section 3.1. The description of Tables 2 and 3 is qualitative, and there are too many configurations and panels to discuss. Do we learn anything in terms fundamental flow processes?

Section 3 presents results. Their discussion is presented in section 4. Flow regimes describe unsteady character of the flow into the gully which can be characterise defining 3 flow regimes.

- Figure 3. The quality is again very poor. Is there a way to simplify all this information, so that the figure conveys the main message?

We improved quality of Figure 3, which make it more readably. Velocity and pressure values are important parameters to characterize flows. We represent now with non dimensional parameters to be extended for other applications.

Other minor comments:

-Lines 21, 24. Formatting of citations.

corrected

- Simulatiions, line 108.

corrected

- Lines 114-116 needs to be rewritten: “The outlet boundary at right (street outlet) was set as free outflow, considering outlet zero gradient for all except for the pressure which dynamic part is specified as zero normal gradient rather than total”

Re-phrased

- Line 118, “hear”

Re-phrased

- Line 119, what does “exchanges of air” mean as a boundary condition? Imposing atmospheric pressure?

The atmosphere boundary at the top is defined for alpha (VOF function) allowing inlet or outlet

- Use of italics. Units and other letters should be converted to regular text in latex, with \text{} or some other command in math mode.

corrected

- Line 124. I’m surprised that, given the flow complexity and presumably large Reynolds number, the numerical solutions converge to a steady state. Is this because the grid is very coarse?

With steady state we mean that the main features of the flow, discharge flow distribution, water depth converge. This was specified in the text.

- Line 128 needs to be rewritten “the water volume maintains relatively stable since the beginning”

corrected

- Line 137. “Particullary”

removed

- Line 150. “Responsable”

corrected

- Lines 162-164. This phrase should be rewritten, as it is difficult to read: “Therefore, in the first seconds a wave is formed increasing water depth downstream, a substantial part of the flow is drained to channel downstream and the flow through the pipe outlet always come from the gully downstream area”

Re-phrased

- Lines 175-177. The use of parenthesis inside another parenthesis is very confusing.

removed

- Lines 185. “Almost similar” is too vague.

Substituted by “show the same trend”

- Lines 185-188, the use of commas seems incorrect.

corrected

- Lines 214-216 should be rewritten (maybe split into two sentences?). - Lines 216-220 should to be rewritten (we need at least a few commas). - Line 230 “…and the location of the pipe outlet.”- Lines 233-245. This reads like a one-sentence paragraph that needs to be split into several sentences. The use of semicolons seems odd.- Lines 247-254, same as for lines 233-245.-  Lines 256-271, same as for lines 233-245.

All the text in 4.1 was re-written

- Figure 4, caption. Maybe “according to each configuration”?

included

Reviewer 3 Report

Brief summary

A numerical investigation based on Volume-of-Fluid technique of the hydraulic behaviour of gullies is presented. Various tests were performed by varying the inflow along with the size (e.g. the diameter) and the location of the outlet pipe at the gully invert. Several plots and tables are shown to analyse pressure and velocity distribution and to describe various flow regimes establishing into the gully.

Broad comments

The manuscript is quite well organized. The results are deeply described. Maybe this section might be slightly shortened. Authors are kindly invited to make the result section more concise. Conclusions are quite appropriate.

Besides technical observations, which are detailed in the following and in the Specific comments section, the main concern of the Reviewer is about the English grammar. A lot of grammatical errors and typos are present and the text comprehension (especially with reference to the description of the flow regimes) results to be very difficult because some concepts are not well written. The English language is definitely not appropriate with reference to the expected level for publishing research papers. Consequently, the Reviewer suggests working deeply to correct the manuscript and improve the English language, even by resorting to the help of English-experts.

The text many typos, as detailed into the specific comments.

Two general comments are reported as follows:

It would be opportune to      present the experimental results (pressure, velocities) by making the      concerned hydraulic parameters not-dimensional. So doing, the wideness of the      manuscript outcomes increases significantly because readers interested in      modelling similar numerical models of gullies can compare their results with      Authors’ ones, even by considering different gully geometries. For this      reason, the Reviewer suggest to adapt plots of Figures 3 and 5 by considering      relative values of pressure, velocities and discharges;

Most of the submitted figures      are out of focus. This hinders the clear readability of the manuscript.      Specific comments, as reported below, will detail the specific figures      which need to be largely improved.

Specific comments

Page 2 lines 7-8: please      improve the English grammar in the sentence “On the other hand, some flow analysis in particular structures and      devices, such as gullies, manholes, search detailed characteristics….”.      Moreover, it would be opportune to use experimental investigations,      for instance, in place of flow analyses;

Page 2 line 18: What do Authors      mean by “local gutter depression”?      Please, revise the use of the word “depression”.      Furthermore, it is convenient to specify here that the “location of the outlet” refers to      the gully outlet;

Page 2 lines 21-22: add systems      after “urban drainage” and them      between “compared” and “with”;

Figure 1: the figure shows an      up-view (please, specify in the figure caption that two up-views are represented)      of two typical types of gullies. Authors should remark this issue, even in      the figure caption. The latter should also specify what the red circles in      the figure indicate;

Page 2 line 31: please substitute      “its” with their;

Page 2 line 34: What do Authors      mean by “unsteady character”?      This is definitely not clear;

Page 3 line 35-37: Why did Authors      specify that gullies were studied under “drainage conditions? Do gullies operate      also under not drainage conditions? Furthermore, please clarify what “large range of flows” means. Were      different flow types investigated? Or Did Authors use “flows” to stand for discharges?

Page 3 lines 39-41: the English      grammar in “Different gully models for different outlet locations and      dimensions were now constructed and simulations for a wide range of inflow      rates are performed” is poor. Please, be careful to the compliance of different      tenses;

Page 3: the sentence within      lines 52 and 57 is excessively long. Please, consider to divide it into two      sentences to make its readability easier. Moreover, it would be convenient      to include momentum and advection equations, as included in the VOF      methods;

Page 4 line 60: please correct      the typo “developped”;

Page 4 line 62: substitute the comma      with a period, after with add Lopes et al (13) thus simulated….;

Page 4 lines 68-69: Authors stated      that their manuscript did not focus on air entrainment into gullies. So,      why did they report details above about the air entrainment model by Lopes      et al (13)?

Page 4 line 81: the modelled      gully is 0:6m x 0:3m x 0:3m. Which is the width? The length? And the      height?

Page 4 lines 87-89: the English      grammar in the sentence “Despite      finer meshing being tested which showed detailed vortex formation, it was      chosen a mesh size that retains the main features of the vortices, while      keeping the calculation time acceptable” is poor. Please improve it;

Page 5 line 91: it is more appropriate      to use outlet pipe and not “pipe      outlet”;

Page 5 lines 90-104: the description      of the 8 gully configurations is lengthy. It may be convenient to insert a      Table where the position and the size of the outlet pipe are reported. The      text would thus result less confused;

Figure 2: the diameter of the      outlet pipe in Figure 2g looks like smaller than that ones in Figs. 2f and      2h. May Authors check id it was drawn wrongly? Moreover, if possible      please improve the resolution of the Figure. As currently presented,      symbols are almost unreadable;

Page 5 lines 107-110: “…with fixed height, hin, and uniform      velocity, Uin = uin, along the depth”. This is not clear. You assumed      uniform flow condition as upstream BC. Why did you report two velocities      Uin and uin? Also the sentence “Channel      slope in simulations was defined by setting gravitational field as g =      (0:00017; 0;-9:81200) and considering xx axis parallel to the channel      bottom, to make it easy to define the mesh” is absolutely unclear. Please,      make these issues clearer. In the end, correct the type “simulatiions” at line 108;

Authors should describe the      outlet boundary conditions by avoiding to refer directly to the OPENFOAM boundary      condition style. What do “zero      gradient” and “zero normal      gradient” stand for? Try to be more generic. For instance: at the outlet      of the outing pipe a hydrostatic pressure was assumed;

Table 1: Did Authors explicate      the meaning of Fr in the text where Table 1 is referred to? Moreover, it      would be useful to add the configurations, for which the various discharges      were run, in the Table;

Page 7 line 124: which flow parameters      did Authors inspect to verify the achievement of steady flow regime? Is it      possible to specify them in the text?

In the first hydraulic      behaviour, was the water volume stable after 15 sec? Or did it continue to      decrease progressively? Specify this issue at page 7 lines 126-127;

Page 7 line 130: What is the “downstream      rising limb”? This is not clear;

Tables from 2 to 6: the legends      showing the velocity field is unreadable;

For flow behaviour 1 (Table 2)      and Q50 a not negligible amount of inflow goes beyond the gully at t = 0.5      s. At this time, the gully results to be drowned especially under      configurations 7 and 8. Do Authors retain to underline this issue in the      main text?

Page 12: in the Table 4, the      third configuration for Q50 is Config. 3 and not “Config. 13”;

Page 13 line 136: Authors      introduced some symbols to denote the main hydraulic parameters into Fig. 2.      They might use these symbols in the main text, consequently. For instance,      use h3 instead of “downstream channel water depth”;

Page 13 line 145: substitute “t ≤ 5 s” with 0.5 < t 5 s;

Page 13 lines 146: Are you sure      that “hydraulic jump” is the correct denomination for identifying the      surface irregularities shown in flow phase 3? Did the flow pass through      the critical state?

Page 14 line 160: Why did      Authors write that “the water depth is      not enough to deflect…”? Maybe you would have underlined that the upstream      water depth (h1) was so large that not all the inflow was deviated in the      outlet pipe;

Tables 5 and 6: Why did you denote      the configurations here as “Config. 6 -1/4”, “Config. 7 -c” and so on,      instead of “Config. 6” and “Config. 7” as previously reported in the main      text and in Tables 2, 3 and 4?

Again, the resolution of plots      needs to be largely. Legends and markers of plots represented in Figure 3      are totally obscure;

Pressure and velocity values      reported in Figure 3 should be relative to respective reference parameters      (uniform flow parameters, for instance) otherwise the experimental values      are hardly to be comparable to other outcomes derived for different gully geometries;

Page 16 and 18 lines 190-195: English      grammar in these sentences is very poor. It needs to be largely improved      otherwise the text comprehension remains rather complicated. Same thought regards      lines 214-217 at page 18;

Page 21 line 280-283: Firstly,      Author miss to describe the meaning of Co and Cw symbols, which are      included in Eqs. (1) and (2). Furthermore, the description of H (uniform      flow depth) is too vague. Where is it defined? At the gully bottom? Or      with reference to the upstream channel invert? Moreover, Authors decided      to denote water depth with the symbol h, as presented in Fig. 2, Why did      they change by using H in Eqs. (1) and (2)? In the end, some confusion may      arise by using both subscripts “o” (orifice) and “0” (upstream). Please,      change one of these subscripts;

Tables 7 and 8: please, add the      dimension of the discharges. Moreover, who is h? It is not highlighted in      the text nor in the Table caption. Please, be careful to the efficiency values      of Table 7 (and consequently of Fig. 5) for D = 80mm, Q10, h = 0:027m.      They seem to be wrong;

How were the discharge      coefficients of Tables 7 and 8 derived? Are they result by fixing Qo and      Qw equal to the Qoutlet_bottom? Please, specify this issue before reporting      Tables 7 and 8.

Author Response

Dear Editor and Reviewer 3,

We thank you for your fruitful comments that clearly improved the paper. An English expert reviewed the paper. Apart from a certain number of commas,  all the changes are in blue.

Please find answers, comments and modifications above each reviewer’s issue.

Comments and Suggestions for Authors

Brief summary

 A numerical investigation based on Volume-of-Fluid technique of the hydraulic behaviour of gullies is presented. Various tests were performed by varying the inflow along with the size (e.g. the diameter) and the location of the outlet pipe at the gully invert. Several plots and tables are shown to analyse pressure and velocity distribution and to describe various flow regimes establishing into the gully.

Broad comments

The manuscript is quite well organized. The results are deeply described. Maybe this section might be slightly shortened. Authors are kindly invited to make the result section more concise. Conclusions are quite appropriate.

Besides technical observations, which are detailed in the following and in the Specific comments section, the main concern of the Reviewer is about the English grammar. A lot of grammatical errors and typos are present and the text comprehension (especially with reference to the description of the flow regimes) results to be very difficult because some concepts are not well written. The English language is definitely not appropriate with reference to the expected level for publishing research papers. Consequently, the Reviewer suggests working deeply to correct the manuscript and improve the English language, even by resorting to the help of English-experts. The text many typos, as detailed into the specific comments.

We made a major revision, we reviewed all the text, we ask for help of an English-expert and we improve the English language. We reduce the number of simulation results and made a more concise results section. Description of flow regimes was re-written.

Two general comments are reported as follows:

It would be opportune to present the experimental results (pressure, velocities) by making the concerned hydraulic parameters not-dimensional. So doing, the wideness of the manuscript outcomes increases significantly because readers interested in modelling similar numerical models of gullies can compare their results with Authors’ ones, even by considering different gully geometries. For this reason, the Reviewer suggest to adapt plots of Figures 3 and 5 by considering relative values of pressure, velocities and discharges;

We present hydraulic parameters not-dimensional in Figure 3 as suggested. We work on both Figures

Most of the submitted figures are out of focus. This hinders the clear readability of the manuscript. Specific comments, as reported below, will detail the specific figures which need to be largely improved.

We reduce number of results presented and improved the quality.

Specific comments

Page 2 lines 7-8: please      improve the English grammar in the sentence “On the other hand, some flow analysis in particular structures and      devices, such as gullies, manholes, search detailed characteristics….”.      Moreover, it would be opportune to use experimental investigations,      for instance, in place of flow analyses;

We rephrased the sentence: On the other hand, some flow analysis in particular structures and devices, such as gullies, and manholes, search detailed characteristics were analysed using either experimental investigation {Chanson2004,PfisterGisonni2014} or CFD, in order to understand flow behaviour and look for parameters that could characterize such flows.

Page 2 line 18: What do Authors mean by “local gutter depression”? Please, revise the use of the word “depression”.      Furthermore, it is convenient to specify here that the “location of the outlet” refers to the gully outlet;

The expression “local gutter depression” was removed and substituted by “geometry”. The local depression describes the gutter slope at the face of the inlet. We specified the gully outlet: Factors such as the local geometry and the location of the gully outlet are also known to influence the flow.

Page 2 lines 21-22: add systems after “urban drainage” and them between “compared” and “with”;

Done: Computational fluid dynamics (CFD) is becoming increasingly utilised as part of simulation schemes for hydraulic structures in urban drainage systems tested different turbulence models and compared them with experimental measurements, …

Figure 1: the figure shows an up-view (please, specify in the figure caption that two up-views are represented) of two typical types of gullies. Authors should remark this issue, even in  the figure caption. The latter should also specify what the red circles in the figure indicate;

Done: \caption{Two up-views View of two different typical gullies with different location of the gully outlet (connection to the pipe system): a) upstream; b) near centre} (red circles indicate the gully outlet position)

Page 2 line 31: please substitute      “its” with their;

Done: The inlet efficiency of gullies is not well known and their discharge …

Page 2 line 34: What do Authors mean by “unsteady character”? This is definitely not clear;

We rephrased the sentence: In that work, the authors found that initial conditions and unsteady character have a great influence in the velocity field and the time to attain steady conditions.

Page 3 line 35-37: Why did Authors specify that gullies were studied under “drainage conditions? Do gullies operate also under not drainage conditions? Furthermore, please clarify what “large range of flows” means. Were different flow types investigated? Or Did Authors use “flows” to stand for discharges?

Sometimes, gullies operate under exceptional situations after the sewer system becomes pressurized (Lopes et al 2015 - doi/full/10.1080/1573062X.2013.831916). Yes, we meant a large range of discharge flows. We clarified the sentence: This work aims to study the hydraulic behaviour of gullies in usual drainage conditions featuring different locations of the vertical gully outlet that connects the gully to the buried drainage system, considering different dimensions and a large range of discharge flows, and 2D and 3D analyses.

Page 3 lines 39-41: the English grammar in “Different gully models for different outlet locations and dimensions were now constructed and simulations for a wide range of inflow rates are performed” is poor. Please, be careful to the compliance of different tenses;

We rephrased the sentence: Different gully models for different outlet locations and dimensions were now constructed and simulations for those were used to simulate a wide range of discharge flows inflow rates are performed analysing the following parameters:

Page 3: the sentence within  lines 52 and 57 is excessively long. Please, consider to divide it into two sentences to make its readability easier. Moreover, it would be convenient to include momentum and advection equations, as included in the VOF methods;

We rephrase the text- Equations were included.

Page 4 line 60: please correct  the typo “developped”;

Done: developed

Page 4 line 62: substitute the comma with a period, after with add Lopes et al (13) thus simulated….;

Done.

Page 4 lines 68-69: Authors stated that their manuscript did not focus on air entrainment into gullies. So, why did they report details above about the air entrainment model by Lopes et al (13)?

Authors presented Lopes et al. model and results, which are validated and show that for the purpose of the present research, air – entrainment model is not a major concern. The details informing Lopes et al. (13) developed an air entrainment model can be removed. However, authors maintained the sentence adding some justification.

Page 4 line 81: the modelled gully is 0:6m x 0:3m x 0:3m. Which is the width? The length? And the      height?

We rephrased the sentence: The computational domain of the flow, aiming to represent a length x width x height = $0.6 m$ x $0.3 m$ x $0.3 m$ gully placed …

Page 4 lines 87-89: the English grammar in the sentence “Despite finer meshing being tested which showed detailed vortex formation, it was chosen a mesh size that retains the main features of the vortices, while keeping the calculation time acceptable” is poor. Please improve it;

We rephrased the sentence: Despite finer meshing being We tested three meshes. The two finer mesheswhich showed equivalent detailed vortex details, therefore formation, it was chosen we chose an intermediate mesh size, which retains the main features of the vortices, while keeping the calculation time acceptable.

Page 5 line 91: it is more appropriate to use outlet pipe and not “pipe outlet”;

Done: outlet pipe was replaced  pipe outlet along the text

Page 5 lines 90-104: the description of the 8 gully configurations is lengthy. It may be convenient to insert a Table where the position and the size of the outlet pipe are reported. The  text would thus result less confused;

We rephrased the Figure 2 presentation and we add a Table as suggested. Table 1 shows outlet pipe characteristics for each configuration.

Figure 2: the diameter of the outlet pipe in Figure 2g looks like smaller than that ones in Figs. 2f and 2h. May Authors check id it was drawn wrongly? Moreover, if possible      please improve the resolution of the Figure. As currently presented,      symbols are almost unreadable;

The Figure 2g was smaller than 2f and 2h. It was corrected

Page 5 lines 107-110: “…with fixed height, hin, and uniform velocity, Uin = uin, along the depth”. This is not clear. You assumed      uniform flow condition as upstream BC. Why did you report two velocities      Uin and uin?

This was clarified: Table 2 (being Table 1 in the 1st submission) presents the inflow condition for the different discharge flows, uniform height, h_{in}, and uniform velocity, U_{in}, which was consider constant along the depth.

Also the sentence “Channel slope in simulations was defined by setting gravitational field as g = (0:00017; 0;-9:81200) and considering xx axis parallel to the channel bottom, to make it easy to define the mesh” is absolutely unclear.

We rephrase the sentence: We considered x-axis parallel to the channel bottom instead of being horizontal. Thus the acceleration due to gravity instead of being consider vertical, was set to g = (0:00017; 0;-9:81200) to consider a 1% slope

Please, make these issues clearer. In the end, correct the type “simulatiions” at line 108;

done

Authors should describe the outlet boundary conditions by avoiding to refer directly to the OPENFOAM boundary  condition style. What do “zero gradient” and “zero normal gradient” stand for? Try to be more generic. For instance: at the outlet of the outing pipe a hydrostatic pressure was assumed;

Done

Table 1: Did Authors explicate  the meaning of Fr in the text where Table 1 is referred to? Moreover, it would be useful to add the configurations, for which the various discharges were run, in the Table;

Fr is now defined before the table (now Table 2) as well as Reynolds number as suggested by Reviewer 2: … Froude and Reynolds numbers are included in the Table.

Page 7 line 124: which flow parameters did Authors inspect to verify the achievement of steady flow regime? Is it  possible to specify them in the text? In the first hydraulic behaviour, was the water volume stable after 15 sec? Or did it continue to decrease progressively? Specify this issue at page 7 lines 126-127;

We specified the main flow properties analysed to consider steady conditions and we clarify. We add this into the text.

Page 7 line 130: What is the “downstream rising limb”? This is not clear;

Rising limb was substituted by water depth

Tables from 2 to 6: the legends  showing the velocity field is unreadable;

Tables 2 to 6 (Now Tables 3 to 7) were reformulated in order to improve quality. We erase some intermediate plots. Just one readable legend is presented valid for all.

For flow behaviour 1 (Table 2) and Q50 a not negligible amount of inflow goes beyond the gully at t = 0.5 s. At this time, the gully results to be drowned especially under  configurations 7 and 8. Do Authors retain to underline this issue in the main text?

Yes, we reformulate the section and we introduce a sentence for Behaviour 1: For Behaviour 1, the gully initially (t= 0.5 s) shows to be drowned especially under configurations 7 and 8 and a considerable amount of inflow goes downstream with large variation of water depth.

Page 12: in the Table 4, the  third configuration for Q50 is Config. 3 and not “Config. 13”;

Corrected

Page 13 line 136: Authors  introduced some symbols to denote the main hydraulic parameters into Fig. 2. They might use these symbols in the main text, consequently. For instance,  use h3 instead of “downstream channel water depth”; Page 13 line 145: substitute “t ≤ 5 s” with 0.5 < t  5 s;

We simplified the figure 2 as requested and h3 was removed, as was not an important parameter. Calculations were done starting at 0. In spite of the first plot we present is t=0.5, this is true for t ≤ 5.

However, this section was reformulated.

Page 13 lines 146: Are you sure  that “hydraulic jump” is the correct denomination for identifying the   surface irregularities shown in flow phase 3? Did the flow pass through  the critical state?

hydraulic jump is the correct denomination. The flow is supercritical upstream and become subcritical downstream.

Page 14 line 160: Why did  Authors write that “the water depth is not enough to deflect…”? Maybe you would have underlined that the upstream      water depth (h1) was so large that not all the inflow was deviated in the      outlet pipe;

This was reformulated

Tables 5 and 6: Why did you denote  the configurations here as “Config. 6 -1/4”, “Config. 7 -c” and so on,      instead of “Config. 6” and “Config. 7” as previously reported in the main      text and in Tables 2, 3 and 4?

Corrected. Just the number of Configuration is mention and Tabe 1 indicate its position

Again, the resolution of plots needs to be largely. Legends and markers of plots represented in Figure 3 are totally obscure;

This was modified

Pressure and velocity values reported in Figure 3 should be relative to respective reference parameters (uniform flow parameters, for instance) otherwise the experimental values are hardly to be comparable to other outcomes derived for different gully geometries;

We made a new no dimensional Figure 3, with better quality and with just one legend, valid for all.

Page 16 and 18 lines 190-195: English grammar in these sentences is very poor. It needs to be largely improved otherwise the text comprehension remains rather complicated. Same thought regards lines 214-217 at page 18;

All work was revised

Page 21 line 280-283: Firstly, Author miss to describe the meaning of Co and Cw symbols, which are included in Eqs. (1) and (2). Furthermore, the description of H (uniform flow depth) is too vague. Where is it defined? At the gully bottom? Or with reference to the upstream channel invert? Moreover, Authors decided to denote water depth with the symbol h, as presented in Fig. 2, Why did they change by using H in Eqs. (1) and (2)? In the end, some confusion may arise by using both subscripts “o” (orifice) and “0” (upstream). Please, change one of these subscripts; Tables 7 and 8: please, add the dimension of the discharges. Moreover, who is h? It is not highlighted in the text nor in the Table caption. Please, be careful to the efficiency values of Table 7 (and consequently of Fig. 5) for D = 80mm, Q10, h = 0:027m. They seem to be wrong; How were the discharge coefficients of Tables 7 and 8 derived? Are they result by fixing Qo and Qw equal to the Qoutlet_bottom? Please, specify this issue before reporting Tables 7 and 8.

We reformulate Equations as we used to calculate discharge coefficients. We detailed calculation details to reach supercritical uniform conditions and to calculate discharge coefficients and efficiency. We call uniform conditions as Hin. Tables 7 and 8 (now 8 and 9) present Qoutletpipe and Qdownstream channel , and the addition of both give Qin.

Round 2

Reviewer 2 Report

The authors have addressed my previous comments, and I recommend publication.

Author Response

Dear Editor and Reviewer 2,

We thank you again for your revision and words.

Reviewer 3 Report

The manuscript has been largely improved and Authors have accomplished all my comments and suggestions. I congratulate them for the good work to revise the manuscript.

One of my main concern was about the English grammar. The quality of English language and style is now undoubtedly improved. I think that a further enhancement might be reached by reading again the manuscript and fixing some remaining poor English sentences.

I have only some short comments which will be fixed by Authors without any particular trouble. My comments are reported as follows:

Page 2 line 8: The meaning of acronym      CFD should be explicated where it appears in the text for the first time;

Page 2 line 21: Please,      substitute system with systems;

Page 2 line 34: despite the      Authors’ reply to my comment, unsteady      character persists in the text without any clear explanation. Please,      clarify what unsteady character stands      for;

Table 1: please, substitute 0.2      (numerical value of Dpo) with 0.20 (two decimal places are required      due to the accuracy in reporting the smaller value of the outlet pipe      diameter (Dpo=0.08 m). Make also sure that the same      correction will be done in all the main text;

Authors have never explicated      the meaning of symbol Ppo, which is for instance included in the      caption of Figure 1 and into Table 1. Please, add the corresponding explanation      in the main text;

Page 7 line 106: Authors here used      Uin as the approach uniform flow velocity, whereas they reported      uin to indicate the same parameter into Table 2. Please,      homogenize the symbols;

Page 8 lines 126-132:      Configurations from 1 to 5 are all characterized by Dpo = 0.08      m. It is thus not useful to repeat the value of Dpo where these      configurations are mentioned. Same observation is valid for configurations      from 6 to 8. Please, make also sure that this modification will be      extended within the overall main text;

Table 4: the caption of this      figure report D, instead of Dpo, to denote the outlet pipe diameter;

Tables 8 and 9 some still      confusing issues persist in these tables and in the text referring to the      tables. Authors should specify that Qin = Qoutlet channel + Qoutlet      bottom (as denoted into Tables 7 and 8). Moreover, Qoutlet bottom for D =      80 mm, Q10, Hin = 0.027 m and config. 1 (Table 8) assumes a very small      value compared with the other ones. Is it correct?

Author Response

Dear Editor and Reviewer 3,

We thank you again for the additional comments that clearly improved the paper.

Please find in blue all the changes between Revision 1 and 2.

Please find answers, comments and modifications above reviewer 3’s issues.

One of my main concern was about the English grammar. The quality of English language and style is now undoubtedly improved. I think that a further enhancement might be reached by reading again the manuscript and fixing some remaining poor English sentences.

We revised again all the text with care.

Page 2 line 8: The meaning of acronym      CFD should be explicated where it appears in the text for the first time;

Done

Page 2 line 21: Please,      substitute system with systems;

Done

Page 2 line 34: despite the      Authors’ reply to my comment, unsteady      character persists in the text without any clear explanation. Please,      clarify what unsteady character stands      for;

We erase “unsteady character ” as it was dispensable. The main reason is the initial conditions. The explanation would require a lot of text and the idea is explained in the results.

Table 1: please, substitute 0.2      (numerical value of Dpo) with 0.20 (two decimal places are required      due to the accuracy in reporting the smaller value of the outlet pipe      diameter (Dpo=0.08 m). Make also sure that the same      correction will be done in all the main text;

We change al for mm units, 80 mm and 200 mm, to be consistent. Apart from the table, we found one more time at line 94.

Authors have never explicated      the meaning of symbol Ppo, which is for instance included in the      caption of Figure 1 and into Table 1. Please, add the corresponding explanation      in the main text;

Ppo was explained in the Figure 2 caption. However we included in the main text and we simplified Figure 2 caption.

Page 7 line 106: Authors here used      Uin as the approach uniform flow velocity, whereas they reported      uin to indicate the same parameter into Table 2. Please,      homogenize the symbols;

Uin  and uin are two different parameters, Uin means uniform flow velocity, average along depth.  uin is a vector of average velocities in each finite volume of the left boundary (inflow), which have the same values of Uin . We change uin to Uin in Table 2 to avoid confusion.

Page 8 lines 126-132:      Configurations from 1 to 5 are all characterized by Dpo = 0.08      m. It is thus not useful to repeat the value of Dpo where these      configurations are mentioned. Same observation is valid for configurations      from 6 to 8. Please, make also sure that this modification will be      extended within the overall main text;

Done, we erase Dpo = 80 mm, where it appeared after Configuration 1 to 5 and Dpo= 200 mm, where it appeared after Configuration 6 to 8.

Table 4: the caption of this      figure report D, instead of Dpo, to denote the outlet pipe diameter;

Done, We also changed po to op since we call outlet pipe instead pipe outlet.

Tables 8 and 9 some still      confusing issues persist in these tables and in the text referring to the      tables. Authors should specify that Qin = Qoutlet channel + Qoutlet      bottom (as denoted into Tables 7 and 8).

Done

Moreover, Qoutlet bottom for D =      80 mm, Q10, Hin = 0.027 m and config. 1 (Table 8) assumes a very small      value compared with the other ones. Is it correct?

Yes, there was an extra zero, that was erased.